# SL-S4Wave: Self-Supervised Learning of Physiological Waveforms with Structured State Space Models

**Feng Wu**                                                                    *wufeng@mit.edu*
*Massachusetts Institute of Technology*

**Harsh Deep**                                                                 *hdeep@mit.edu*
*Massachusetts Institute of Technology*

**Eric Lehman**                                                                *lehmer16@mit.edu*
*OpenEvidence, USA*

**Sanyam Kapoor**                                                              *sanyam@nyu.edu*
*New York University*

**Guoshuai Zhao**                                                  *guoshuai.zhao@xjtu.edu.cn*
*Xi'an Jiaotong University*

**Rahul Krishnan**                                               *rahulgk@cs.toronto.edu*
*University of Toronto*

**Gari Clifford**                                                  *gari.clifford@emory.edu*
*Emory University*

**Li-wei H Lehman**                                                *lilehman@mit.edu*
*Massachusetts Institute of Technology*

**Reviewed on OpenReview:** *https://openreview.net/forum?id=km0xS3jZeO*

## Abstract

Modeling long-sequence medical time series data, such as electrocardiograms (ECG), poses significant challenges due to high sampling rates, multichannel signal complexity, inherent noise, and limited labeled data. While recent self-supervised learning (SSL) methods, based on various encoder architectures such as convolutional neural networks, have been proposed to learn representations from unlabeled data, they often fall short in capturing long-range dependencies and noise-invariant features. Structured state space models (S4) excel at long-sequence modeling, but existing S4 architectures fail to capture the unique characteristics of multichannel physiological waveforms. In this work, we propose SL-S4Wave, a self-supervised learning framework that combines contrastive learning with a tailored encoder built on structured state space models. The encoder incorporates multi-layer global convolution using multiscale subkernels, enabling the capture of both fine-grained local patterns and long-range temporal dependencies in noisy, high-resolution multichannel waveforms. Extensive experiments on real-world datasets demonstrate that SL-S4Wave (1) consistently outperforms state-of-the-art supervised and self-supervised baselines in a challenging arrhythmia detection task, (2) achieves high performance with significantly fewer labeled examples, showcasing strong label efficiency, and (3) maintains robust performance on long waveform segments, highlighting its capacity to model complex temporal dynamics in long sequences that most existing approaches fail to efficiently model, and (4) transfers effectively to unseen arrhythmia types, underscoring its robust cross-domain generalization. We additionally evaluate SL-S4Wave on multiple EEG tasks, achieving superior performance over strong baselines, demonstrating generalizability of our approach beyond cardiac waveforms.

# 1 Introduction

Monitoring patients in clinical settings involves continuous acquisition of physiological signals through sensors. These high-resolution waveforms, such as Electrocardiograms (ECG) and Electroencephalogram (EEG), provide clinicians with crucial insights into patient state and disease progression. However, analyzing such signals automatically remains challenging. Supervised learning methods require large volumes of annotated data, yet high-quality medical labels are both scarce and costly to obtain. At the same time, physiological waveforms are noisy, multichannel, and exhibit complex temporal dynamics, making them difficult to model effectively.

Self-supervised learning (SSL) has recently emerged as a promising direction for representation learning from unlabeled medical waveforms (Zhou et al., 2022; Kiyasseh et al., 2021; Lalam et al., 2023). By pretraining on large datasets and fine-tuning on limited labeled samples, SSL methods reduce reliance on annotation and have shown encouraging results. Contrastive learning, as a classic SSL method, can obtain effective data representations in a completely unlabeled setting (Hu et al., 2024). For ECG and EEG signals, contrastive learning is also a relatively mature self-supervised learning method (Kiyasseh et al., 2021; Ma et al., 2026). However, most existing approaches are designed for short waveform segments and encoder architectures such as convolutional networks, which excel at capturing local patterns but struggle with long-range temporal dependencies (Birnbaum et al., 2019; Li et al., 2022). This limitation is especially critical in patient monitoring, where clinically important events such as arrhythmias need to be detected from long-term monitoring data. Some Transformer-based methods can extract information from long-term data, but due to the quadratic complexity of the attention mechanism increasing with length, they require substantial resources to compute long-range dependencies (Alghieth, 2025). Moreover, existing SSL methods typically do not explicitly address robustness to noise, which is pervasive in bedside monitoring data.

Meanwhile, advances in structured sequence models, such as Structured State Space Models (S4) (Gu et al., 2021), have demonstrated strong capabilities for modeling very long sequences in language and speech. S4 leverages state-space dynamics and convolutional reformulations to efficiently capture dependencies across thousands of time steps. Extensions such as Structured Global Convolution (SGConv) (Li et al., 2022) have further improved scalability through multi-scale temporal modeling. However, these models have not yet been tailored to the unique challenges of physiological signals, which require both fine-grained local feature extraction and robust modeling of multichannel, noisy, long-duration waveforms.

To address these gaps, we propose **SL-S4Wave**, a self-supervised learning framework designed specifically for long-sequence, multi-channel physiological waveform modeling. Our approach contains two key components. First, we propose **S4Wave**, a structured state space deep-learning model that extends global convolution architectures with multiscale kernels, residual connections, and cross-channel modeling, enabling effective representation of both short-term and long-range temporal dynamics in high-resolution multi-channel physiological waveforms. We used residual connections to mitigate vanishing gradients in deep networks and gating mechanism to control the information flow between layers. Second, we present a **self-supervised pretraining framework** based on the S4Wave-encoder with contrastive objectives, combining invariance to noise perturbations with temporal coherence, enabling the model to learn robust and generalizable representations from unlabeled physiological waveforms.

We evaluate SL-S4Wave on multiple physiological signal modalities spanning two clinical domains. In the main text, we focus on arrhythmia detection in critical care using ECG, arterial blood pressure (ABP), and photoplethysmography (PPG) signals, particularly the challenging ventricular tachycardia (VT) alarm classification task (Drew et al., 2014; Zhou et al., 2022; Clifford et al., 2015). We additionally validate our approach on EEG-based state recognition across three tasks, with results presented in the Appendix F. Pretrained using unlabeled waveform data and fine-tuned on labeled data with expert-verified annotations, SL-S4Wave consistently outperforms state-of-the-art supervised and self-supervised baselines. The model exhibits strong cross-domain generalization across arrhythmia types, high label efficiency under limited supervision, and robust performance on extended waveform segments, demonstrating its capacity to model long-range temporal dependencies in physiological time series.

Our main contributions are as follows:

**The S4Wave encoder for multivariate physiological waveforms.** We introduce S4Wave, a deep encoder architecture that adapts structured state-space models to multichannel physiological waveforms. Its global convolution kernels span the entire input sequence, enabling direct modeling of long-range temporal dependencies. Combined with residual connections and gating mechanisms, S4Wave captures both local dynamics and long-range temporal structure in multivariate waveform data.

**The SL-S4Wave self-supervised learning framework.** Building on the S4Wave encoder, we develop SL-S4Wave, a self-supervised framework with contrastive learning objectives tailored to physiological signals, learning noise-robust representations while capturing long-range temporal context.

**Long-sequence representation learning.** We show that SL-S4Wave scales effectively to long input sequences, consistently outperforming all baselines as sequence length increases. In contrast, CNN-based methods degrade on longer sequences, highlighting the advantage of our approach for physiological waveform tasks that require extended temporal context.

**Cross-domain transferability.** We demonstrate that representations learned by SL-S4Wave transfer effectively to arrhythmia types that are unseen during pretraining. The results show that SL-S4Wave learns robust and generalizable signal representations, maintaining strong accuracy even when downstream label sets are limited or substantially differ from the pretraining task. We further evaluate SL-S4Wave on three EEG benchmarks and observe similarly strong performance (results in Appendix F).

We released the SL-S4Wave implementation and the pretrained model weights publicly available on GitHub [1].

## 2 Related Work

**Structured State Space for Sequence Modeling** Recent works have focused on enhancing classic State-Space Models (SSM) to more efficiently model sequential data using deep learning. For example, Rangapuram et al. (2018) used recurrent neural networks to learn parameters in SSM. However, the most significant progress came from Gu et al. (2021) with the introduction of the structured state space model (S4), which reduces the computational complexity of SSM in modeling long sequences using a special state transition matrix (Gu et al., 2020). These low-rank and normal matrices enable SSM to compute global convolution kernels efficiently through fast Fourier transform across the entire sequence. Subsequently, some works have further improved the shortcomings of S4 in areas such as model architecture (Smith et al., 2022) and convolution (Raghu et al., 2023), and have started applying it to tasks such as natural language processing (Dao et al., 2023) and time series analysis (Zhou et al., 2023; Ma et al., 2026).

**Self-Supervised Learning (SSL) in Time Series.** Self-supervised learning (SSL) enables models to extract informative representations from unlabeled data by generating surrogate supervision signals. SSL has achieved remarkable success across modalities such as images, videos, and physiological signals, and its adoption in biomedical time series is rapidly growing. In clinical contexts, SSL has been applied to ECG and EEG signals (Zhou et al., 2022; Kiyasseh et al., 2021; Raghu et al., 2023; Wang et al., 2023), structured physiological data (McMaster et al., 2023), and medical imaging (Rivail et al., 2019; Eldele et al., 2021). Recent advances in SSL for time series have introduced more general architectures that improve temporal representation learning beyond conventional CNNs and RNNs. For example, Fraikin et al. (2024) proposed T-Rep, a transformer-based framework that learns time-aware representations via temporal embeddings, while Xu et al. (2024) introduced a retrieval-augmented reconstruction strategy to enhance contrastive time-series learning.

Most prior SSL methods for physiological time series continue to rely on convolutional or recurrent architectures, which struggle to model long-range dependencies due to limited receptive fields and gradient instability—thereby constraining pretraining to short input windows (typically 2–4 seconds) (Raghu et al., 2023; Lan et al., 2022; Kiyasseh et al., 2021). This limitation hampers the capture of global temporal patterns that extend across multiple physiological cycles. In contrast, the proposed **SL-S4Wave** framework

---

[1] https://github.com/ML-Health/SLS4Wave

performs contrastive learning over substantially longer waveform segments, enabling the model to capture richer temporal dependencies and improve representation quality.

**Representation Learning in Physiological Waveforms.** Learning effective representations from physiological waveforms is challenging, particularly when labeled data are scarce. Lehman et al. (2018) demonstrated that incorporating domain priors—such as beat-level cardiac structure—can substantially reduce input length (from 10 to 3 seconds) to achieve competitive performance, highlighting the value of physiological knowledge in label-efficient learning. In contrast, SL-S4Wave obviates the need for cardiac beat detection, and directly learns effective representations from long segments of multi-channel waveforms. Empirical studies further show that CNN-based models often outperform RNNs and Transformers for arrhythmia detection and related tasks (Zhou et al., 2022; Lehman et al., 2024), with 1-D CNNs excelling on smaller datasets and deeper fully convolutional networks (FCNs) achieving superior performance on larger ones. Our results are consistent with these prior findings, but demonstrated that the proposed SL-S4Wave is consistently more effective in the arrhythmia detection task than these previous convolution neural network baselines, particularly when the labeled dataset is sparse. More recently, ECGFounder (Li et al., 2025) introduced a large-scale foundation model trained on over 10 million labeled ECGs spanning 150 diagnostic categories, reaching expert-level accuracy and strong transferability. In contrast, **SL-S4Wave** focuses on learning robust and generalizable representations from noisy, unlabeled multichannel waveforms, capturing long-range temporal dependencies without reliance on large annotated corpora.

## 3 Methodology

In this section, we present SL-S4Wave, a self-supervised learning framework for long-sequence, multivariate physiological waveform modeling, depicted in Figure 1. SL-S4Wave integrates two core components. First, we develop S4Wave, a structured state space model (SSM)–based encoder that enriches global convolution operations with multiscale kernels, residual connections, and cross-channel interactions, enabling effective representation of both short-term morphology and long-range dependencies in high-resolution physiological signals. Second, we introduce a self-supervised contrastive pretraining strategy built upon the S4Wave encoder, which learns robust representations that are invariant to noise and maintain temporal consistency, allowing the model to leverage large volumes of unlabeled physiological data to learn robust and transferable latent representations.

### 3.1 Preliminary

We define the input multivariate physiological signals as $X \in \mathbb{R}^{C \times L}$, where L is the sequence length of each trajectory, $C$ is the number of channels, such as ECG and arterial blood pressure (ABP). As the number of channels $C$ increases, the amount of information in physiological signals also grows, and modeling cross-channel interactions becomes increasingly challenging. Our task is considered as a two-stage process, consisting of a pre-training stage and a fine-tuning stage. In the pre-training stage, we need to train an encoder $f_{enc}$ using an unlabeled dataset $D_{pt} = \{x_n^{pt}\}_{n=1}^{N_{pt}}$ of size $N_{pt}$ which is composed of unlabeled samples $x^{pt}$ to obtain high-dimensional representations $h_n^{pt} = f_{enc}(x_n^{pt})$ for different samples. In the fine-tuning stage, we use the encoder $f_{enc}$ trained in the pre-training stage as a feature extraction module to train the fine-tuning dataset containing labels $D_{ft} = \{x_n^{ft}, y_n^{ft}\}_{n=1}^{N_{ft}}$ with size $N_{ft}$, where $x_n^{ft}$ denotes fine-tuning samples and $y_n^{ft}$ denotes the corresponding labels. Then, we use a classifier $C_\psi$ to process the representations $h_n^{ft}$ obtained from the encoder $f_{enc}$ and obtain the final results $y_\theta^{ft} = C_\psi(h_n^{ft})$. In this work, our main focus is on the performance of the encoder, while the classifier is constructed using a Multi-Layer Perceptron (MLP).

### 3.2 Obtaining Efficient Representations with S4Wave

**S4Wave signal encoder** During pretraining, the model architecture used in our framework differs from traditional state-space models (SSMs), which are typically designed to model underlying physical dynamical processes. (For background information on the SSM model and theory, please refer to the Appendix A.) In contrast, our formulation leverages the SSM structure as a powerful sequence encoder that maps physiological

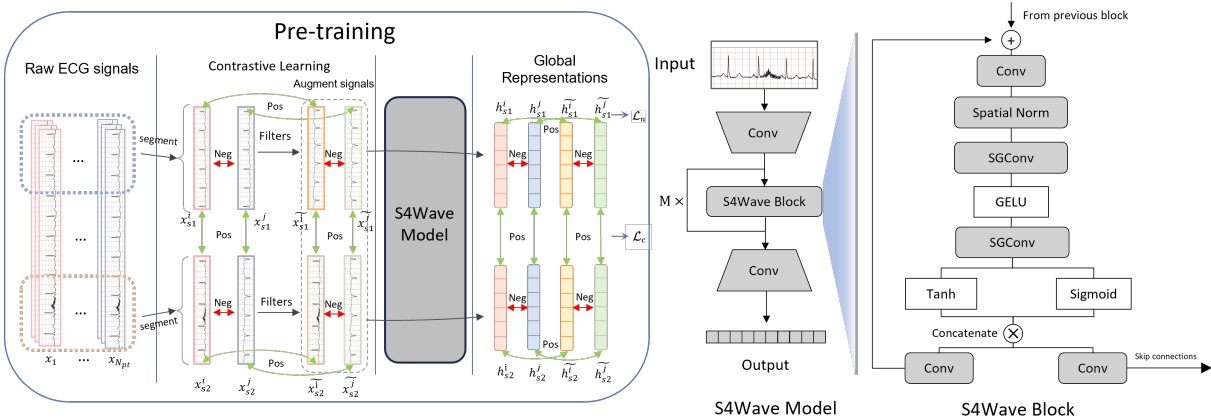

Figure 1: Overview of proposed Self-Supervised Learning Structured State Space Model (SL-S4Wave). Left: During pretraining, waveform segments from different patients form negative pairs, while each filtered segment and its corresponding original segment form a positive pair. Right: The S4Wave model learns expressive latent representations of waveform data through $M$ stacked S4Wave blocks. Skip connections between blocks help mitigate vanishing gradients and facilitate stable optimization.

waveforms into a latent feature space. To this end, we introduce **SL-S4Wave**, which employs an S4Wave block to learn robust representations from multichannel physiological signals capturing both local and long-range temporal structure. Specifically, for the augmented physiological signals $X \in \mathbb{R}^{C \times L}$, we utilize a Conv1D layer to embed the input to hidden space $h_{in} \in \mathbb{R}^{H \times L}$, $H$ is the number of hidden dimensions:

$$h_{in} = Conv1D(X). \tag{1}$$

Then, we define an S4Wave block that is built on a ResNet structure with a skip connection. In each block, the input is first added to the residual output from the previous block, $\mu_r^{m-1}$, where $m \in \{1, 2...M\}$ denotes the current block and $M$ denotes the total number of blocks, and then normalized before being passed into the next module:

$$h_{SN} = SN(Conv1D(h_{in} + \mu_r^{m-1})), \tag{2}$$

where $SN(\cdot)$ is the spatial normalization layer for normalizing the channel-level data Deng et al. (2021). As the number of channels increases, normalizing along the channel dimension helps the model converge faster and train more stably. $h_{SN}$ is the hidden representation after the $SN(\cdot)$ layer. In S4Wave, we use SGConv as the implementation of our SSM layer. SGConv is a structured SSM model using convolution architecture, and its convolution structure can be represented as an FFT formula:

$$y_{fft} = F_N^{-1} D_k F_N \mu_{fft}, \qquad D_k = \text{diag}(\overline{K} F_N), \tag{3}$$

where $\mu_{fft}$ and $y_{fft}$ are the input signal and output signal of the FFT, and $F_N$ denotes the Discrete Fourier Transform (DFT) matrix of size $N$. This FFT convolution has a computational complexity of $O(nlog(n))$. We use two layers of SSM (i.e. SGConv here) to process the input, and between these two layers, we use the GELU function as an activation function, which can enhance the non-linear characteristics of the network. The SGConv convolution kernel $\overline{K}$ in Eq.4 can thus be initialized as:

$$\overline{K} = \frac{1}{Z} concat[k_0, k_1, ..., k_{\lfloor log_2(\frac{L}{d}) \rfloor + 1}], \qquad h_{ssm} = \overline{K}_2 \text{GELU}(\overline{K}_1 h_{SN}), \tag{4}$$

where the $i$ th subkernel $k_i = \alpha^i \text{Upsample}_{2^{max[i-1,0]}d}(w_i)$, and kernel weight $w_i \in \mathbb{R}^d$ is a learnable parameter for the $i$th convolution subkernel. The number of subkernels can be calculated as $\lfloor log_2(\frac{L}{d}) \rfloor + 1$, where $d$ denotes the first subkernel size, and $L$ denotes sequence length. The upsampling function $\text{Upsample}(x)_{2^{max[i-1,0]}d}$ denotes upsampling $w_i$ from length $d$ to length $2^{max[i-1,0]}d$, where for the first subkernel size $d$. The entire kernel $\overline{K}$ is concatenated as $concat[d, 2d, 4d, ..., 2^{i-1}d]$ and the subscript $2^{max[i-1,0]}d$ indicates the size of the current $i$-th subkernel. $Z$ denotes a normalization constant and $\alpha$ denotes a decay

coefficient. $\overline{K}_1$ and $\overline{K}_2$ are two different sets of convolutional kernels based on Equation 4. Due to the introduction of decay coefficients, upsampling, and normalization parameters, SGConv is easier to compute and more efficient compared to the original S4 (Gu et al., 2021). In the Appendix C, we explore how decay helps SGConv obtain better representations on long time series. Through the SSM, we obtain a $\mathbb{R}^{H \times L} \to \mathbb{R}^{H \times L}$ mapping. Then we used a gating unit to obtain a part of the output of the entire S4Wave block and the input to the next block. We divide $h_{ssm}$ into two parts $h_{ssm}^1$ and $h_{ssm}^2$ along the $H$ dimension.

$$h_{gate} = tanh(W_f \circledast h_{ssm}^1) \odot \sigma(W_g \circledast h_{ssm}^2), \qquad h_{ssm} = concat[h_{ssm}^1, h_{ssm}^2] \tag{5}$$

$$\mu^r = Conv1D(h_{gate}), s^o = Conv1D(h_{gate}), \tag{6}$$

where $tanh(\cdot)$ and $\sigma(\cdot)$ are tanh function and sigmoid function, $\odot$ denotes an element-wise multiplication operator, $W_f$ and $W_g$ are learnable convolution filters. We use $\circledast$ to denote the 1D convolution operation throughout this section. $\mu^r$ becomes the residual part input to the next block, while $s^o$ will be aggregated to the output of SSM blocks through a skip connection. Finally, we will use a convolutional layer to map the features $concat[s_1^o, s_2^o, ...s_M^o]$ to representations $z_o \in \mathbb{R}^{H \times L}$, which will be the input to the Classifier head of the downstream task.

$$z_o = Conv1D(Concat[s_1^o, s_2^o, ...s_M^o]) \tag{7}$$

### 3.3 Contrastive Learning on Physiological Waveforms

**Noise-resilient loss.** Physiological waveforms from bed-side monitors are often corrupted by sensor artifacts, motion interference, and environmental noise, which can degrade representation quality and robustness of the model. We introduce a noise-resilience contrastive loss to minimize the impact of noise on the model. For each sampled input segment, we apply signal-processing filters to generate a noise-reduced version as an augmented waveform segment. The original waveform and its filtered counterpart are treated as a positive pair, i.e. two views of the same latent signal. During the pre-training process, we aim to reduce the distance $\langle h; \tilde{h} \rangle$ between the representation of the original data $h$ and the representation after noise reduction through filters $\tilde{h}$, to ensure the model's noise resistance. We describe the specific noise reduction strategies in Appendix D. To meet the form of contrastive learning loss, we choose samples from different patients within the same batch as negative samples $h_j$, which are not filtered so that the model can focus on optimizing the impact of noise. In particular, our positive pair is $\langle h_i; \tilde{h}_i \rangle$, and the negative pair is $\langle h_i; h_j \rangle$. We adopt the cosine similarity between two representations a and b:

$$sim(a, b) = \frac{a^\top b}{\|a\| \, \|b\|} \tag{8}$$

The noise-resilient loss $\mathcal{L}_n$ for sample $i$ is defined as:

$$\mathcal{L}_n = -\frac{1}{N} \sum_{i=1}^{N} \ln \left( \frac{\exp(sim(h_i, \tilde{h}_i)/\tau)}{\sum_{j=1}^{2N} \mathbb{1}_{[i \neq j]} \exp(sim(h_i, h_j)/\tau)} \right) \tag{9}$$

where $N$ is the batch size, $sim(\cdot)$ is the similarity between sample pairs, and here we use cosine similarity. $\tau$ is a temperature hyperparameter. $\mathbb{1}_{[i \neq j]}$ is an indicator function, which means that similarity is calculated when $i \neq j$, and it is 0 if $i = j$.

**Context consistency loss.** In our framework, the contrastive loss is designed to encourage temporal consistency within the same physiological record. Positive pairs are constructed from temporally adjacent segments of a given waveform, while negative pairs are drawn from segments belonging to different waveform records. This setup minimizes the distance between latent representations of nearby segments from the same record, enabling the model to capture coherent morphological patterns over time. Concretely, if we denote two non-overlapping segments of length $L$ seconds sampled from successive intervals within the same record as $h_{s1}^i$ and $h_{s2}^i$, these form a positive pair $\langle h_{s1}^i, h_{s2}^i \rangle$. In contrast, negative pairs are constructed by matching a segment from one record with segments of the same length drawn from different records within the training batch, e.g., $\langle h_{s1}^i, h_{s1}^j \rangle$ where $i \neq j$. This contrastive objective promotes robustness by encouraging invariance

to local temporal variations while ensuring separation between unrelated waveform dynamics. The context consistency loss $\mathcal{L}_c$ is defined as:

$$\mathcal{L}_c = -\frac{1}{N} \sum_{i=1}^{N} \ln \left( \frac{\exp(\text{sim}(h_{s1}^i, h_{s2}^i)/\tau)}{\sum_{j=1}^{2N} \mathbb{1}_{[i \neq j]} \exp(\text{sim}(h_{s1}^i, h_{s1}^j)/\tau)} \right) \tag{10}$$

**Overall pre-training loss.** The overall pre-training loss is composed of the two losses mentioned above: Noise-resilient loss is used to reduce the impact of common noise on the model, and context consistency loss is used to learn temporal dependencies and abnormal patterns in existing data. The losses used during pre-training are:

$$\mathcal{L}_{PT} = \mathcal{L}_n + \lambda \mathcal{L}_c \tag{11}$$

where $\lambda$ is a hyperparameter to balance two losses.

**Fine-tune loss** In the fine-tuning stage, we already have a trained encoder and some labeled data, and the choice of the loss function depends on the specific task. Since our downstream task is alarm discrimination, we use the Binary Cross Entropy (BCE) loss to supervise the fine-tuning stage.

$$\mathcal{L}_{\text{BCE}} = -\frac{1}{N} \sum_{i=1}^{N} [v_i \ln(p_i) + (1 - v_i) \ln(1 - p_i)] \tag{12}$$

where $v_i$ is the true label (0 or 1) of sample $i$, $p_i$ is the predicted probability of the positive class for sample $i$. During the finetuning process, we do not freeze the parameters of the pre-trained encoder. Instead, we continue supervised learning with a smaller learning rate, simultaneously training the parameters of the encoder and the classifier.

## 4 Experiments

### 4.1 Datasets and Tasks

**Finetune tasks** False arrhythmia alarms in intensive care units are a continuing problem. Since arrhythmia alarms can be triggered by factors other than actual medical conditions, such as mechanical malfunctions, patient movement, and misdiagnoses, these false alarms create significant pressure for healthcare providers. Specifically, for a given physiological signal $X \in \mathbb{R}^{C \times L}$, corresponding labeled data $y$ indicate whether the segment of the signal represents a true alarm or a false alarm. We need to train a classifier $C_\theta$ to determine if the alarm is triggered by a true arrhythmia event, given the observed physiological waveforms of each patient prior to the alarm onset. For the EEG tasks, we tested the model's performance on three different subtasks, please see Appendix F for details.

**Datasets** We evaluate our technique using three publicly-available datasets on PhysioNet:

**MIMIC II Arrhythmia dataset.** The arrhythmia database from Multi-Parameter Intelligent Monitoring for Intensive Care (MIMIC II) (Saeed et al., 2011; Goldberger et al., 2000; Pollard et al., 2026) includes more than 8,000 waveform records containing multiparameter physiologic waveforms with human annotations (Aboukhalil et al., 2008). Waveforms were stored at 125 Hz with 8 bit resolution.

**Ventricular Tachycardia annotated alarms from ICUs (VTaC) dataset.** The VTaC (Lehman et al., 2024; Goldberger et al., 2000; Pollard et al., 2026) data set consists of 18,472 VT alarm events from 2,383 patient waveform records collected from multiple ICUs in three major US hospitals. Each waveform recording contains six-minute segments of ECG leads and one or more pulsatile waveforms (photoplethysmogram and/or arterial blood pressure waveforms) sampled at 250 Hz. The final labeled data for fine-tuning consists of 5,037 VT alarms, each annotated by at least two human experts as either true or false alarms.

**PhysioNet Challenge 2015.** The PhysioNet/Computing in Cardiology Challenge 2015 (Clifford et al., 2015; Goldberger et al., 2000) provides a publicly-available dataset with 750 records for algorithm development. Each record contained an alarm for one arrhythmia event and the triggered alarm was reviewed and

labeled by a team of expert annotators to either true or false. The dataset includes five different types of cardiac arrhythmia alarms. Detailed data processing methods and the utilized channels are presented in the Appendix E.

**EEG Datasets.** For the EEG experiments, we pre-trained SL-S4Wave on a large-scale EEG corpus (Obeid & Picone, 2016) and evaluate its performance on three downstream datasets spanning distinct tasks. Details of these datasets are in Appendix F.

## 4.2 Experimental Setup

**Baselines** We compare the performance of the following baseline and models: 1) **Rule-Based Method:** For the rule-based approach, we used the implementation from (Plesinger et al., 2015), a winning entry in the PhysioNet 2015 challenge for false arrhythmia alarm reduction. 2) **FCN**: we use a Fully convolutional neural network as a feature extractor. 3) **SimCLR**: A classic comparative learning model (Chen et al., 2020). Since Kiyasseh et al. (2021) released a heart-signal–adapted version of SimCLR, we adopt their implementation as our SimCLR baseline. 4) **CLOCS (CMSC)** : The self-supervised learning method based on heart signals proposed by Kiyasseh et al. (2021) encourages channel and spatial representations to be similar. We use the implementation of spatial representation similarity (**CMSC**). 5) **TS-TCC**: A contrastive model Eldele et al. (2021) combines cross-view prediction and contrastive learning tasks by creating two views of the raw time series data using weak and strong augmentations. 6) **TF-C**: A novel time-frequency consistency architecture (Zhang et al., 2022) and optimizes time-based and frequency-based representations of the same example to be close to each other. 7) **TS2Vec**: TS2Vec (Yue et al., 2022) uses sampling strategies and hierarchical losses to maintain context invariance across multiple time resolutions. 8) **ECG-JEPA**: ECG-JEPA (Weimann & Conrad, 2025) is a pre-trained ECG classification model based on a visual Transformer. Its pre-training method involves predicting latent features for the ECG model. 9) **ECGFounder**: ECGFounder (Li et al., 2025) is a CNN-based model that leverages real-world ECG annotations from cardiology experts to broaden the diagnostic capabilities of ECG analysis. We place the **implementation details** of all methods in the Appendix D.

**Evaluation.** In addition to common evaluation metrics such as TPR/TNR, Accuracy, F1, Positive Predictive Value and AUC, we use the **Challenge Score** as an important metric as defined by the PhysioNet Challenge 2015 Clifford et al. (2015): $Score = \frac{TP+TN}{TP+TN+FP+5*FN}$.

## 4.3 Results

**Performance in VT Detection across Three Fine-Tuning Datasets** We evaluate our method for detecting true ventricular tachycardia (VT) alarms on three datasets—Challenge 2015-VT, MIMIC II-VT, and VTaC using 10-second waveform segments preceding each alarm as input. Table 1 reports the results. Across nearly all metrics, SL-S4Wave consistently surpasses both supervised and self-supervised baselines. On the small Challenge 2015-VT cohort, SL-S4Wave achieves the highest F1 score (81.84), PPV (75.52), and AUC (91.98), demonstrating strong performance in low-label settings. On the larger MIMIC II-VT dataset, it achieved a Challenge Score of 69.57 and an AUC of 88.56, outperforming all supervised methods (e.g., FCN Score 63.01, AUC 85.87) and every self-supervised competitor. On VTaC, SL-S4Wave attains the top Challenge Score of 83.25 and the highest AUC of 96.01, exceeding the best supervised baseline (FCN, Score 80.83, AUC 94.93). Other self-supervised models also perform competitively on VTaC, but fall short of SL-S4Wave on AUC and Challenge Score. ECGFounder and ECG-JEPA underperformed likely due to the fact that both models are optimized based on 12-lead ECG, whereas the VTac and MIMIC II-VT datasets only contain two leads of ECG, while including ABP and PLETH channels. The models lack optimization for these two channels. These results highlight SL-S4Wave's ability to generalize across datasets of widely varying size and distribution while effectively leveraging self-supervised pretraining. We attribute these gains to the stronger feature extraction capabilities of the S4Wave encoder compared to the CNN-based backbones employed by baselines like SimCLR and CMSC. While standard SSL methods leverage "prior" knowledge, their capacity is often constrained by the local receptive fields of convolutional architectures, which may fail to capture global temporal dynamics in high-sampling-rate physiological signals. Unlike Temporal Convolutional Networks (TCNs) Lea et al. (2016) or dilated convolutional architectures, which

| Datasets | Method | | TPR | TNR | Score | F1 | PPV | AUC |
|---|---|---|---|---|---|---|---|---|
| | Rule-based | | 76.47 | 85.18 | 67.81 | 68.42 | 61.9 | - |
| | Supervised | FCN | $97.65 \pm 2.88$ | $28.52 \pm 4.16$ | $44.17 \pm 4.27$ | $46.03 \pm 2.09$ | $30.12 \pm 1.56$ | $68.69 \pm 4.47$ |
| | | Transformer | $66.67 \pm 0.00$ | $81.82 \pm 14.08$ | $61.11 \pm 8.61$ | $59.43 \pm 10.39$ | $55.71 \pm 15.03$ | $76.97 \pm 12.80$ |
| Challenge'15 | Self-supervised | SimCLR | $90.59 \pm 4.71$ | $25.56 \pm 5.90$ | $37.68 \pm 2.08$ | $42.44 \pm 0.76$ | $27.74 \pm 0.76$ | $56.60 \pm 4.73$ |
| | | CMSC | $89.41 \pm 2.35$ | $28.89 \pm 2.77$ | $39.42 \pm 2.48$ | $43.08 \pm 1.33$ | $28.38 \pm 0.99$ | $59.13 \pm 3.03$ |
| N=343 | | TS-TCC | $80.00 \pm 6.00$ | $60.74 \pm 4.60$ | $54.97 \pm 3.64$ | $52.54 \pm 3.03$ | $39.21 \pm 2.72$ | $75.19 \pm 1.15$ |
| | | TF-C | $67.06 \pm 7.06$ | $49.63 \pm 16.62$ | $40.60 \pm 6.34$ | $41.68 \pm 3.84$ | $31.03 \pm 5.81$ | $63.14 \pm 3.97$ |
| | | TS2Vec | $88.24 \pm 6.44$ | $36.30 \pm 5.93$ | $43.82 \pm 2.48$ | $45.17 \pm 1.64$ | $30.41 \pm 1.22$ | $69.52 \pm 4.04$ |
| | | ECG-JEPA | $57.28 \pm 6.10$ | $71.15 \pm 4.61$ | $44.52 \pm 2.56$ | $50.75 \pm 3.01$ | $45.83 \pm 3.17$ | $74.27 \pm 0.41$ |
| | | ECGFounder | $67.27 \pm 3.80$ | $71.92 \pm 3.75$ | $50.84 \pm 3.01$ | $57.62 \pm 2.99$ | $50.49 \pm 3.48$ | $77.62 \pm 2.30$ |
| | | SL-S4Wave(Ours) | $89.41 \pm 2.35$ | $90.37 \pm 4.29$ | $\mathbf{81.84 \pm 1.69}$ | $\mathbf{81.56 \pm 4.09}$ | $\mathbf{75.52 \pm 8.04}$ | $\mathbf{91.98 \pm 0.33}$ |
| | Rule-based | | 85.77 | 48.74 | 52.37 | 72.59 | 62.92 | - |
| | Supervised | FCN | $92.41 \pm 1.72$ | $52.54 \pm 4.00$ | $63.01 \pm 2.98$ | $77.25 \pm 1.59$ | $66.45 \pm 2.08$ | $85.87 \pm 1.00$ |
| | | Transformer | $97.09 \pm 1.74$ | $11.83 \pm 4.95$ | $51.68 \pm 1.24$ | $68.30 \pm 0.62$ | $52.70 \pm 1.04$ | $62.54 \pm 2.29$ |
| MIMIC II-VT | Self-supervised | SimCLR | $97.38 \pm 1.55$ | $17.56 \pm 6.48$ | $54.79 \pm 3.42$ | $69.82 \pm 2.70$ | $54.33 \pm 3.13$ | $71.42 \pm 4.41$ |
| | | CMSC | $96.67 \pm 2.71$ | $13.48 \pm 8.63$ | $51.82 \pm 2.78$ | $68.49 \pm 1.83$ | $52.93 \pm 1.59$ | $74.41 \pm 2.14$ |
| N=2802 | | TS-TCC | $95.11 \pm 1.45$ | $34.34 \pm 4.35$ | $59.09 \pm 1.92$ | $73.14 \pm 1.00$ | $59.32 \pm 1.43$ | $69.76 \pm 1.67$ |
| | | TF-C | $88.30 \pm 5.36$ | $49.10 \pm 11.28$ | $55.70 \pm 5.07$ | $73.99 \pm 4.03$ | $63.52 \pm 6.41$ | $79.94 \pm 5.22$ |
| | | TS2Vec | $94.11 \pm 3.81$ | $11.33 \pm 5.88$ | $47.47 \pm 2.60$ | $66.77 \pm 0.70$ | $51.68 \pm 0.81$ | $67.31 \pm 2.02$ |
| | | ECG-JEPA | $82.20 \pm 3.93$ | $78.28 \pm 3.34$ | $59.32 \pm 4.90$ | $80.69 \pm 2.40$ | $79.32 \pm 2.54$ | $87.88 \pm 3.08$ |
| | | ECGFounder | $53.74 \pm 6.56$ | $95.51 \pm 1.82$ | $10.16 \pm 3.33$ | $53.06 \pm 1.27$ | $67.16 \pm 0.67$ | $51.81 \pm 0.72$ |
| | | SL-S4Wave(Ours) | $94.37 \pm 0.77$ | $60.21 \pm 2.02$ | $\mathbf{69.57 \pm 1.18}$ | $\mathbf{80.83 \pm 0.59}$ | $\mathbf{70.86 \pm 0.90}$ | $\mathbf{88.56 \pm 0.36}$ |
| | Rule-based | | 94.16 | 62.90 | 67.32 | 65.48 | 50.19 | - |
| | Supervised | FCN | $91.83 \pm 1.31$ | $86.96 \pm 2.60$ | $80.83 \pm 1.65$ | $81.79 \pm 2.15$ | $73.82 \pm 3.70$ | $94.93 \pm 0.38$ |
| | | Transformer | $83.36 \pm 2.39$ | $67.36 \pm 4.51$ | $60.45 \pm 0.88$ | $62.84 \pm 1.54$ | $50.55 \pm 2.93$ | $84.39 \pm 0.86$ |
| VTaC | Self-supervised | SimCLR | $95.91 \pm 0.40$ | $79.01 \pm 2.18$ | $80.10 \pm 1.53$ | $77.14 \pm 1.68$ | $64.23 \pm 2.31$ | $93.73 \pm 0.07$ |
| | | CMSC | $89.49 \pm 2.71$ | $80.23 \pm 8.63$ | $74.03 \pm 2.78$ | $74.81 \pm 1.83$ | $63.96 \pm 1.59$ | $91.85 \pm 2.14$ |
| N=5037 | | TS-TCC | $86.72 \pm 6.68$ | $56.00 \pm 6.33$ | $56.34 \pm 2.72$ | $58.30 \pm 1.59$ | $43.90 \pm 2.14$ | $74.95 \pm 1.38$ |
| | | TF-C | $82.77 \pm 6.66$ | $70.15 \pm 9.01$ | $61.68 \pm 2.21$ | $64.37 \pm 2.80$ | $52.96 \pm 5.68$ | $85.67 \pm 1.54$ |
| | | TS2Vec | $88.32 \pm 4.22$ | $77.39 \pm 3.29$ | $71.23 \pm 5.29$ | $72.06 \pm 3.86$ | $60.61 \pm 3.96$ | $90.64 \pm 1.58$ |
| | | ECG-JEPA | $76.35 \pm 10.15$ | $91.19 \pm 2.17$ | $69.08 \pm 7.44$ | $76.65 \pm 4.49$ | $77.72 \pm 2.31$ | $92.62 \pm 1.46$ |
| | | ECGFounder | $77.92 \pm 6.73$ | $90.65 \pm 1.04$ | $69.84 \pm 5.92$ | $77.26 \pm 4.22$ | $76.76 \pm 2.53$ | $92.51 \pm 3.41$ |
| | | SL-S4Wave(Ours) | $93.98 \pm 1.55$ | $86.96 \pm 1.99$ | $\mathbf{83.25 \pm 0.44}$ | $\mathbf{82.89 \pm 0.97}$ | $73.88 \pm 2.41$ | $\mathbf{96.01 \pm 0.29}$ |

Table 1: **Ventricular tachycardia detection task**: Evaluation results on Challenge 2015-VT, MIMIC II-VT and VTaC dataset, using 10s segment waveforms prior to alarms as input for classification. All self-supervised methods are pre-trained on the VTaC unlabeled dataset. Reported values are the mean and standard deviation over 5 runs (except for the rule-based method, which is deterministic). Result with the highest average values of F1, Score, PPV and AUC in bold. Score = Challenge Score.

extend receptive fields through stacked dilation, the SGConv module in S4Wave employs global convolution kernels whose length matches the entire input sequence, representing a more aggressive design for capturing long-range dependencies. Furthermore, on such small datasets, supervised models often struggle to converge or suffer from severe overfitting because of insufficient training examples. In contrast, SL-S4Wave utilizes effective features learned from the larger unlabeled pre-training dataset to obtain a robust initialization, significantly enhancing performance where labeled data is scarce.

**Ablation study** To validate the effectiveness of each component in our framework, we conducted a comprehensive ablation study on both the pre-training objectives and the S4Wave encoder architecture. The results on the MIMIC II-VT and VTaC datasets are summarized in Table 2. As shown in the first section of Table 2, removing the pre-training stage (w/o Pre) leads to the most significant performance degradation across both datasets (e.g., a 15.99% drop in Score on MIMIC II-VT), confirming that self-supervised pre-training learns transferable features critical for the downstream task. When analyzing specific losses, the model without context consistency loss ($\mathcal{L}_c$) suffers a substantial drop, particularly on the VTaC dataset (Score $\downarrow 8.57\%$), indicating that capturing long-range temporal coherence is essential for identifying arrhythmia patterns. Similarly, removing the noise-resilient loss ($\mathcal{L}_n$) also impairs performance, demonstrating that explicitly modeling invariance to signal artifacts improves robustness against real-world ICU noise.

we further ablated key components within the S4Wave block: the Structured Global Convolution (SGConv), the Gating mechanism, the Spatial Normalization (Snorm), and the GELU activation.

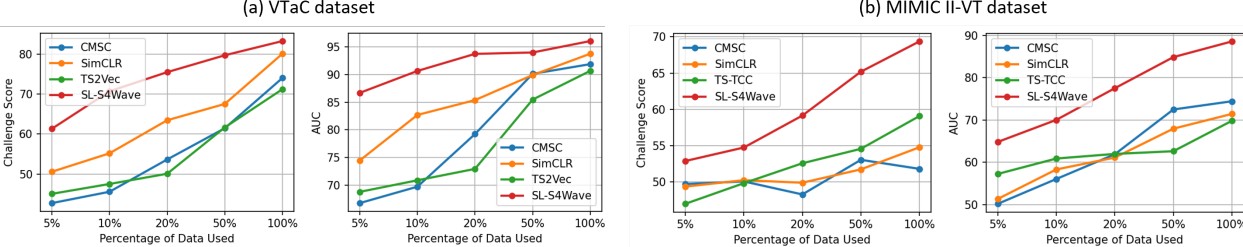

Figure 2: **Fine-Tuning Data Efficiency.** We plot the fine-tuning performance vs. the fraction of training data used for fine-tuning on the VTaC ($N = 5,037$) and MIMIC II-VT ($N = 2,802$) benchmarks and report average of 5 seeds. Even with only 10% of the labeled data, SL-S4Wave matches or surpasses competing self-supervised methods in both Challenge Score and AUC, and continues to scale more favorably as additional data become available. All models used 10s segment of waveform prior to alarm onset as input for classification.

**SGConv:** Replacing the global convolution with standard local convolutions (w/o SGConv) results in the most drastic performance decline among all architectural ablations, with the Challenge Score plummeting by **17.66%** on MIMIC II-VT and **25.96%** on VTaC. This strongly supports our hypothesis that capturing long-range dependencies via global convolution is the core driver of the model's superior performance.

**Gating Mechanism:** Removing the gating unit (w/o Gate) causes a 15.75% drop in Score on MIMIC II-VT. This suggests that the gating mechanism effectively regulates information flow, allowing the model to selectively retain relevant temporal features while filtering out irrelevant dynamics.

**Normalization and Activation:** Both Spatial Normalization (w/o Snorm) and GELU activation (w/o GELU) prove to be indispensable. For instance, removing Snorm leads to an 11.97% score reduction on MIMIC II-VT, highlighting the importance of addressing cross-channel distribution shifts in multivariate physiological signals.

| Method | MIMIC II-VT | | | | VTaC | | | |
|---|---|---|---|---|---|---|---|---|
| | **Score** | $\Delta\%$ | **AUC** | $\Delta\%$ | **Score** | $\Delta\%$ | **AUC** | $\Delta\%$ |
| *Ablation on Contrastive Loss* | | | | | | | | |
| w/o Pre | 58.44 | ↓15.99% | 81.57 | ↓7.90% | 77.84 | ↓6.51% | 94.20 | ↓1.88% |
| w/o $\mathcal{L}_c$ | 65.41 | ↓5.99% | 82.50 | ↓6.84% | 76.12 | ↓8.57% | 94.11 | ↓1.98% |
| w/o $\mathcal{L}_n$ | 66.46 | ↓4.47% | 87.69 | ↓0.99% | 80.40 | ↓3.42% | 94.67 | ↓1.39% |
| *Ablation on Model Architecture* | | | | | | | | |
| w/o GELU | 62.45 | ↓10.23% | 84.42 | ↓4.67% | 79.12 | ↓4.96% | 94.70 | ↓1.36% |
| w/o Gate | 58.61 | ↓15.75% | 80.92 | ↓8.63% | 78.82 | ↓5.32% | 94.46 | ↓1.61% |
| w/o SGConv | 57.28 | ↓17.66% | 78.27 | ↓11.62% | 61.64 | ↓25.96% | 84.81 | ↓11.67% |
| w/o Snorm | 61.24 | ↓11.97% | 81.33 | ↓8.16% | 79.96 | ↓3.95% | 94.60 | ↓1.47% |
| **SL-S4Wave (Full)** | **69.57** | – | **88.56** | – | **83.25** | – | **96.01** | – |

Table 2: **Ablation study on SL-S4Wave using different contrastive losses and model architecture.** The "w/o $\mathcal{L}_c$" setting refers to using self-supervised learning without the context consistency loss. The "w/o $\mathcal{L}_n$" setting is without the noise-resilient loss. The "w/o Pre" refers to using S4Wave for direct supervised training on labeled dataset without any pre-training. All experiments conducted using 10s segments.

**Labeled Data Efficiency** We evaluate the effectiveness of pre-trained representations by fine-tuning models using fractions of labeled data (5%, 10%, 20%, 50%, and 100%) from the VTaC (N=5,037) and MIMIC II-VT (N=2,802) datasets (see Figure 2). Across both datasets and all labeled data sizes, SL-S4Wave consistently achieves the best Challenge Score and AUC, particularly excelling in the low-data regime. Notably, with only 5–10% of labeled data, SL-S4Wave significantly outperforms other contrastive

learning baselines, highlighting its ability to generalize with minimal supervision. These results underscore the benefits of self-supervised pretraining—especially for physiological waveform classification tasks with limited labeled data. All models were trained using 10-second waveform segments preceding alarm onset.

| | Dataset | Method | Score | F1 | PPV | AUC |
|---|---|---|---|---|---|---|
| Pre-trained | VTaC | SL-S4 | $67.76 \pm 2.46$ | $71.20 \pm 0.01$ | $62.59 \pm 0.56$ | $90.58 \pm 0.01$ |
| | | SL-SGConv | $60.43 \pm 0.44$ | $65.32 \pm 0.97$ | $57.70 \pm 2.41$ | $84.44 \pm 0.29$ |
| | | SL-S4Wave | $\mathbf{83.25 \pm 0.44}$ | $\mathbf{82.89 \pm 0.97}$ | $\mathbf{73.88 \pm 2.41}$ | $\mathbf{96.01 \pm 0.29}$ |
| | MIMIC-II | SL-S4 | $56.70 \pm 0.77$ | $73.92 \pm 0.48$ | $62.75 \pm 0.89$ | $79.62 \pm 0.81$ |
| | | SL-SGConv | $48.93 \pm 3.62$ | $68.25 \pm 1.23$ | $54.65 \pm 0.69$ | $67.22 \pm 1.91$ |
| | | SL-S4Wave | $\mathbf{69.57 \pm 1.18}$ | $\mathbf{80.83 \pm 0.59}$ | $\mathbf{70.86 \pm 0.90}$ | $\mathbf{88.56 \pm 0.36}$ |
| No Pre-train | VTaC | S4 | $56.70 \pm 5.23$ | $60.76 \pm 4.53$ | $50.89 \pm 4.80$ | $79.96 \pm 5.56$ |
| | | SGConv | $68.38 \pm 0.84$ | $71.54 \pm 0.59$ | $62.51 \pm 1.07$ | $88.62 \pm 0.17$ |
| | | S4Wave | $\mathbf{77.74 \pm 2.75}$ | $\mathbf{76.97 \pm 4.43}$ | $\mathbf{69.70 \pm 9.03}$ | $\mathbf{91.08 \pm 2.84}$ |
| | MIMIC-II | S4 | $50.99 \pm 0.76$ | $52.30 \pm 1.02$ | $67.54 \pm 0.39$ | $64.52 \pm 1.80$ |
| | | SGConv | $54.69 \pm 0.50$ | $54.69 \pm 0.50$ | $70.69 \pm 0.19$ | $56.41 \pm 0.49$ |
| | | S4Wave | $\mathbf{58.44 \pm 2.04}$ | $\mathbf{74.71 \pm 1.08}$ | $\mathbf{63.39 \pm 2.88}$ | $\mathbf{81.57 \pm 1.00}$ |

Table 3: SL-S4Wave outperforms prior S4-based encoders (specifically the original S4 and SGConv) with and without pretraining across different datasets. All experiments use 10s segment waveforms prior to alarms as input for classification. No pre-train refers to supervised training directly on labeled data only without pre-training on unlabeled data.

**SL-S4Wave Outperforms Prior S4-based Encoders with and without Pretraining**  To evaluate the effectiveness of the proposed S4Wave encoder, we compare its performance with prior S4-based encoder architectures. Table 3 compares S4Wave with prior S4-based approaches. Our results indicate that SL-S4Wave outperforms prior S4-based encoders with and without self-supervised pretraining. To evaluate the effectiveness of the proposed S4Wave encoder, we compare SL-S4Wave with the original S4 and SGConv, both with and without self-supervised pretraining, across the VTaC and MIMIC-II VT datasets. In the self-supervised setting, we replace the S4Wave block with the original S4 or SGConv modules while keeping the same loss, denoted as SL-S4 and SL-SGConv, respectively. As shown in Table 3, SL-S4Wave consistently surpasses these baselines: on VTaC it achieves an AUC of 96.01, a 5.43 percentage-point gain (or 5.99% increase in AUC) over SL-S4, and on MIMIC-II it reaches an AUC of 88.56, exceeding S4 by 8.94 percentage points (or 11.22% increase in AUC). Importantly, the Challenge Score metric improves significantly, with SL-S4Wave improving the pre-trained VTaC score by more than 22% and the MIMIC-II score by nearly 23% relative to the best S4-based baseline.

Even without pretraining, the S4Wave architecture demonstrates clear advantages. When trained from scratch, S4Wave outperforms both S4 and SGConv across all metrics. For instance, on VTaC, S4Wave achieved an an AUC of 91.08 (surpassing AUC of 88.62 by SGConv), while on MIMIC II, its AUC 81.57 surpasses S4 by more than 26%. On VTaC, S4Wave achieves a Challenge Score of 77.74, over 13% improvement over the S4-encoder with the best Challenge Score (with SGConv Challenge Score 68.38), while on MIMIC-II it reaches 58.44, representing an improvement of roughly 15% compared with SGConv. Interestingly, SGConv performs slightly worse with pretraining than without, whereas our S4Wave consistently benefits from pretraining. These findings highlight both the strength of the proposed S4Wave block in capturing long-range dependencies and complex temporal patterns in multi-channel physiological waveforms.

**Performance Comparison of S4Wave with other Deep Learning Encoder Architecture**  To further evaluate the effectiveness of the proposed S4Wave encoder, we compare its performance with other deep learning encoder architectures, including FCN, ResNet and Transformers. In Appendix B.3, results from Table 6 demonstrate that the proposed S4Wave encoder achieved superior performance with and without self-supervised pre-training compared to representative deep learning baselines, including **convolutional** (FCN, ResNet) and **attention-based** (Transformer) architectures, highlighting its ability to model long-range temporal dependencies more effectively.

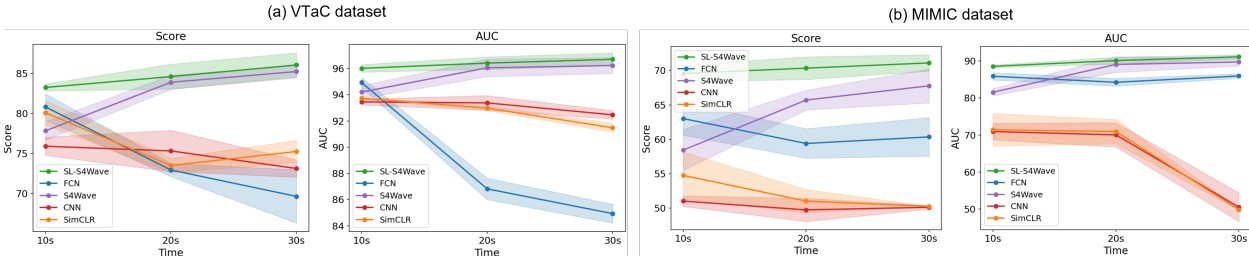

Figure 3: Results using different sequence lengths with SL-S4Wave and baseline model in the two dataset. All experiment conducted with 5 different seeds and reported their mean and standard deviation. The performance of our proposed SL-S4Wave model enhanced further with longer sequences of time series data, significantly outperforming the baseline FCN with contrastive learning. Waveforms were resampled at 125 Hz. Each 30-second segment corresponds to 3,750 time steps. S4Wave is supervised learning without pretraining.

**Performance of SL-S4Wave on Longer Sequences.** ANSI/AAMI EC13 cardiac-monitoring standards require that alarms trigger within 10 s of arrhythmia onset, and most prior machine-learning studies therefore limit model inputs to the 10 s preceding an alarm. We hypothesized, however, that capturing longer pre-alarm context could improve detection.

In this section, we evaluate whether our model could achieve better results over a longer time frame. Figure 3 shows the performance of different models on varying input lengths. (See Tables Tables 9 and 10 in the Appendix for more detailed performance numbers.) Our results show that SL-S4Wave benefits substantially from longer input windows. On VTaC, extending the segment length from 10 s to 30 s raises the Challenge Score from **83.25** to **86.06** and the AUC from **96.01** to **96.70**, while maintaining high F1 (82.9 → 84.1) and PPV (73.8 → 74.6). Similarly, on MIMIC II-VT the Challenge Score improves from **69.57** to **71.09** and the AUC from **88.56** to **91.14** as the window increases from 10 s to 30 s. These results confirm that the model effectively leverages additional temporal context.

In contrast, convolution-based baselines degrade with longer inputs. For example, on VTaC the FCN's Challenge Score drops from 80.83 at 10 s to 69.66 at 30 s, and its AUC falls from 94.93 to 89.71; the CNN shows a similar decline. SimCLR's self-supervised encoder also weakens substantially (Score 80.10 → 75.26, AUC 93.73 → 91.48). The reduction in performance for these approaches may be due to the limited receptive field of fixed-kernel convolutions, which hampers robust modeling of long sequences. Even without self-supervised pre-training, the supervised S4Wave encoder benefits from longer windows. On VTaC, its Challenge Score rises from 77.84 at 10 s to 85.24 at 30 s, with AUC improving from 94.20 to 96.45, demonstrating strong representation learning for extended sequences. Overall, SL-S4Wave not only surpasses all baselines at the standard 10-second input but also maintains and even amplifies its advantage when modeling up to 30 s of pre-alarm data, underscoring its ability to capture long-range temporal dependencies while meeting clinical alarm-timing requirements. These results show that the proposed SL-S4Wave architecture can effectively exploit long-range waveform dependencies, and self-supervised pre-training further enhances this ability, allowing SL-S4Wave to improve accuracy even with three-times-longer input sequences.

**Transferability to Other Downstream Tasks** We assessed cross-domain generalization using the PhysioNet Challenge 2015 and MIMIC-II Arrhythmia datasets, each containing five arrhythmia types (ASY, EBR, ETC, VFB, VTA). All self-supervised methods, including SL-S4Wave, were pretrained solely on unlabeled VT alarms. Figure 4 plots results on the MIMIC II dataset. Table 5 in Appendix reports performance on the MIMIC dataset. On the MIMIC II Arrhythmia dataset, SL-S4Wave achieves the highest performance on nearly all arrhythmia types, with the exception of EBR. Notably, despite being pretrained exclusively on unlabeled VT data, most self-supervised models, including SL-S4Wave, perform well on VTA alarms. However, most other approaches' performance drops when evaluated on other arrhythmia types with highly imbalanced distributions (e.g., ASY) or limited sample sizes (e.g., VFB). SL-S4Wave consistently shows

more robust generalization across these challenges, highlighting its superior transferability. For example, SL-S4Wave achieves an AUC of 92.17 and 96.53 on ASY and VFB respectively, over 6% improvement over the next best self-supervised model. Similarly, on the Challenge 2015 dataset, SL-S4Wave outperforms all other deep learning models on all arrhythmia alarms in AUCs. It achieves the highest Challenge Score in all arrhythmia types, except in ETC. These findings suggest that SL-S4Wave learns generalized and discriminative representations that extend effectively to previously unseen alarm types, even when trained on a single type during pretraining. Overall, these results demonstrate that SL-S4Wave learns robust, discriminative representations that transfer across arrhythmia types, maintaining strong accuracy even when the fine-tuning labels are scarce or the target rhythm differs markedly from the pretraining task.

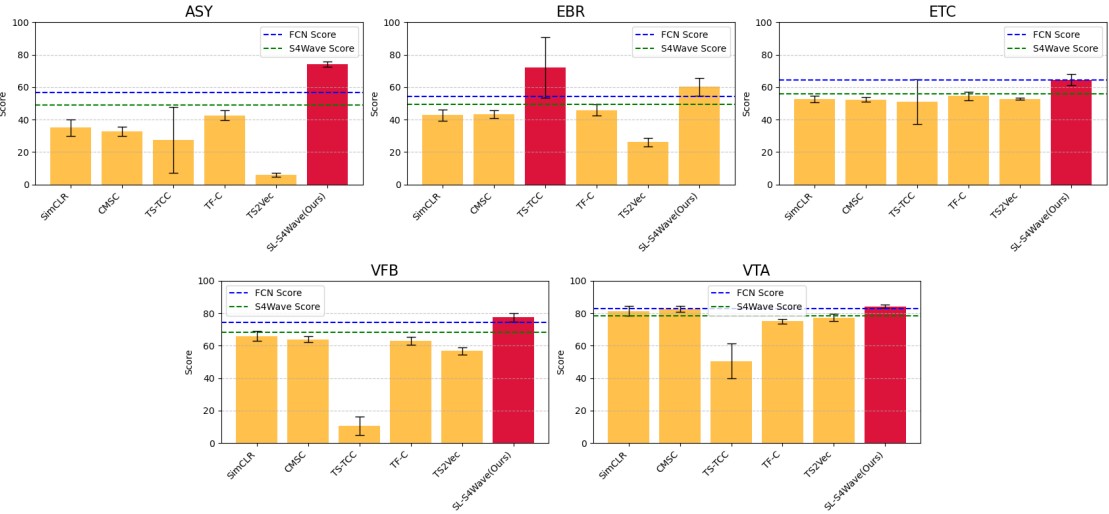

Figure 4: **Broad transferability of SL-S4Wave.** We further show that SL-S4Wave provides a strong pretrained model for adaptation to a wide variety of datasets related to various types of arrhythmia. This figure shows the 5 different arrhythmia results in MIMIC dataset. In all cases, SL-S4Wave tends to at least match or exceed the performance (as measured by Challenge Score) of the best method trained from scratch. Detailed results are in Appendix B.1

## 5 Discussions and Conclusion

In this work, we introduced SL-S4Wave, a self-supervised learning framework built upon the proposed S4Wave encoder for robust representation learning from physiological waveforms. S4Wave effectively models both local and long-range temporal dependencies across multichannel signals and scales to substantially longer waveform segments than conventional convolutional or attention-based architectures. This capability results in richer, more temporally coherent waveform representations. Our evaluation on real-world ECG datasets demonstrates that SL-S4Wave consistently outperforms strong supervised and self-supervised baselines, including convolutional (e.g., FCN, ResNet) and attention-based (e.g., Transformer) models, across multiple arrhythmia alarm validation tasks. Notably, the framework exhibits high label efficiency, achieving competitive performance even with very limited annotations, and demonstrates robust cross-dataset generalization to related arrhythmia classification and EEG tasks. These results underscore the value of combining structured state-space sequence modeling with contrastive self-supervision for biomedical time-series representation learning. Future work will integrate uncertainty quantification for improved reliability in safety-critical settings. We will also investigate foundation model pretraining for structured state-space models for a broader range of clinical tasks.

### Acknowledgments

This research was funded by NIH grants R01EB030362, R01HL181348, and R21HL177773. L Lehman was in part supported by a grant of the Korea Health Technology R&D Project through the Korea Health Industry

Development Institute (KHIDI), funded by the Ministry of Health & Welfare, Republic of Korea (grant number: RS-2024-00403047).

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

# A Structured State-Space Model

The state-space model is a classic model in control theory, and it represents the operational state of a system using first-order differential equations (ODE). A continuous state-space model can be defined in the following form:

$$\begin{aligned} x'(t) &= Ax(t) + Bu(t), \\ y(t) &= Cx'(t) + Du(t), \end{aligned} \tag{13}$$

where $u(t)$ is a vector that represents the input of the system, while the $y(t)$ is a vector that represents the output of the system. $x(t)$ represents the latent state of the system, and its derivative $x'(t)$ w.r.t time defines how the latent representation evolves over time based on both the current state and the input. $A, B, C, D$ here are the state, input, output and feedforward matrices, defining the relationship between the input, output and state vector. To apply the model in a discrete space, we need to discretize the SSM as follows:

$$\begin{aligned} x_k &= \bar{A}x_{k-1} + \bar{B}u_k, y_k = \bar{C}x_k + \bar{D}u_k \\ \bar{A} &= (I - \Delta/2 \cdot A)^{-1}(I + \Delta/2 \cdot A) \\ \bar{B} &= (I - \Delta/2 \cdot A)\Delta B, \end{aligned} \tag{14}$$

where $\Delta$ is a trainable time-step size parameter. Following Gu et al. (2021), $\bar{D}$ is set equal to 0 since it can be replaced by the residual connection. Now, Eq.2 resembles an architecture similar to RNN, allowing us to recurrently compute $x_k$. Let the initial state be $x_{k-1} = 0$, and we can unroll Eq.2 as follows:

$$y_k = \overline{CA^kB}u_0 + \overline{CA^{k-1}B}u_1 + ... + \overline{CAB}u_{k-1} + \overline{CB}u_k \tag{15}$$

$$y = \overline{K}u, \quad \overline{K} = (\overline{CB}, \overline{CA^1B}, ..., \overline{CA^kB}). \tag{16}$$

where $y$ is output sequence $y = (y_0, y_1, ..., y_k)$, while the $u$ is the input sequence $u = (u_0, u_1, ..., u_k)$. And $\overline{K}$ is defined as a SSM convolution kernel. Therefore, SSM can be transformed from the form of a recurrent neural network to a convolutional neural network, allowing us to use the fast Fourier transform to efficiently compute the SSM convolutional kernel $\overline{K}$. In fact, the main computational cost of SSM lies in the $A$ matrix. Since the $A$ matrix is a learnable parameter matrix, it means that every parameter update requires computing the very long convolutional kernel $\overline{K}$. Based on the effective training method proposed by Gu et al. (2020), which introduces a HIPPO matrix to efficiently initialize the matrix A. By decomposing the state transition matrix A into a low-rank matrix and a skew-symmetric matrix, the computational cost of the convolutional kernel is effectively reduced.

# B Additional Results

## B.1 Transferability Evaluation: Challenge 2015 Dataset - Performance on Cross-Domain Data

In the main text, we only report results of MIMIC II Arrhythmia and VTaC datasets. In this section, we present the detailed classification results of various models on the Challenge 2015 dataset (see Table. 4). Note that in this experiment, we integrated all arrhythmias into one dataset and performed partitioning on this dataset. Therefore, the results for VT and VTA in the table differ from those of the main experiment. The SL-S4Wave model achieved the best performance on most metrics in five different arrhythmia tasks. We highlight the following results.

- **SL-S4Wave with pre-train out-performed the best self-supervised baseline in average Challenge Scores in four out of the five arrhythmia types.**

- **SL-S4Wave with pre-training significantly out-performed the supervised approach FCN in AUC across all four arrhythmia types.**

Although TS2Vec was able to outperform the SL-S4Wave model in the extreme tachycardia (ETC) task in Score and F1, but it significantly under-performed in other types of arrhythmia. We also observe that many models exhibit high variance in performance across runs. This variability is largely due to the limited size of the PhysioNet 2015 dataset—particularly the VFB subset, which contains only 40 training samples—making model optimization inherently unstable. Despite this, our method achieves consistently lower standard deviations in many settings. Even when using pre-trained deep learning models, effective training remains difficult under such sparse supervision, especially in cross-domain evaluations. Nevertheless, several self-supervised approaches still outperform fully supervised baselines (e.g., FCN), indicating that pre-training provides useful prior knowledge that improves learning under severe data scarcity.

## B.2 Full Results of Transferability Evaluation on MIMIC II Dataset

Table 5 shows the results of different baseline methods on the MIMIC II arrthymia dataset. The experiment in MIMIC II also follows the experimental setup of B.1, using data from 5 arrhythmias for training. Therefore, the VTA results will differ from those of the main experiment MIMIC-VT. Here are some highlighted results.

- **In the ASY, ETC, VFB, and VTA scenarios, the SL-S4Wave model outperforms other self-supervised learning models by 4%-76% on the Challenge Score metric. There is also an improvement of 6%-23% in the AUC metric.**

- **SL-S4Wave with pre-training significantly out-perform the supervised model FCN in average Challenge Score by 2%-32% in ASY, EBR, VFB and VTA, while achieving similar performance with FCN in ETC.**

In the EBR, SL-S4Wave achieved the second best Challenge Score, after TS-TCC. While TS-TCC performed better than SL-S4Wave and other approaches in the EBR task, it significantly under-performed in all other arrhythmia types. In addition, we observed that some self-supervised models perform well on specific subsets of data, while others do not. For example, TS-TCC performs very well on EBR but poorly on VFB; CMSC performs poorly on ASY and EBR but well on VFB and VTA. This may be due to the different positive pairs of various self-supervised methods and the different focuses learned during the pre-training phase. Our method benefits from sampling from longer intervals and learns more universal representations.

## B.3 Effect of Encoder Architecture and Pre-training in SL-S4Wave

In this section, we compare the performance gap between SL-S4Wave and baseline models with and without pre-training. We train FCN, ResNet He et al. (2015), and Transformer Vaswani et al. (2017) as encoder models within the pipeline. Detailed results are presented in Table 6.

Table 6 reports the downstream classification performance of SL-S4Wave and three alternative encoder architectures—FCN, ResNet18, and Transformer—trained with and without self-supervised pre-training on two datasets (VTaC and MIMIC II). The Transformer used in this experiment is the standard implementation of PyTorch. Its parameters are: d model is 512, head is 8, feed forward is 2048, and there are 6 layers.

In the VTaC dataset, the pre-trained SL-S4Wave model achieves the highest AUC and Challenge Score metrics. The pre-trained FCN has a higher AUC but a lower Score, which may be due to the fact that pre-training enhances the model's ability to judge negative samples, but as a trade-off, there is a certain decrease in the model's ability to judge positive samples. This also happens on the SL-S4Wave. This might be explained by the fact that the unlabelled data may contain more positive samples than negative ones. Meanwhile, the ResNet has significant performance improvements after pre-training. This demonstrates that our model can learn efficient representations during self-supervised learning, enhancing downstream classifiers in better accomplishing their tasks. On the relatively smaller MIMIC II dataset, all baseline models have shown improvement in the two important metrics AUC and Challenge Score. The pre-trained SL-S4Wave consistently outperformed the non-pre-trained SL-S4Wave model. Transformer also continued this trend. On the VTaC dataset, the pre-trained Transformer's Score and AUC metrics greatly exceeded those of the original Transformer, while on the MIMIC dataset there was a slight improvement. This suggests that self-supervised learning performs better on smaller datasets.

| Datasets | Method | | TPR | TNR | Score | F1 | PPV | AUC |
|---|---|---|---|---|---|---|---|---|
| ASY

N=120 | Supervised | FCN | 96.00 ± 8.00 | 20.95 ± 2.33 | 34.56 ± 4.31 | 36.35 ± 2.86 | 22.42 ± 1.76 | 54.10 ± 9.36 |
| | | S4Wave | 100.00 ± 0.00 | 77.14 ± 0.00 | 81.54 ± 0.00 | 67.96 ± 0.00 | 51.67 ± 0.00 | 88.57 ± 0.00 |
| | Self-supervised | SimCLR | 96.00 ± 8.00 | 19.05 ± 0.00 | 33.03 ± 3.18 | 35.78 ± 2.51 | 21.99 ± 1.47 | 53.14 ± 1.64 |
| | | CMSC | 100.00 ± 0.00 | 19.05 ± 5.55 | 37.04 ± 4.92 | 37.04 ± 4.92 | 22.73 ± 5.44 | 61.52 ± 2.78 |
| | | TS-TCC | 32.00 ± 16.00 | 59.05 ± 18.71 | 36.38 ± 14.37 | 23.31 ± 14.88 | 18.79 ± 13.25 | 46.19 ± 17.75 |
| | | TF-C | 60.00 ± 17.89 | 36.19 ± 14.00 | 32.46 ± 14.12 | 28.97 ± 11.23 | 19.25 ± 8.25 | 46.67 ± 12.65 |
| | | TS2Vec | 88.00 ± 9.80 | 23.81 ± 5.22 | 33.18 ± 2.96 | 34.57 ± 2.29 | 21.53 ± 1.27 | 47.81 ± 8.33 |
| | | ECG-JEPA | 65.00 ± 5.74 | 20.00 ± 44.72 | 26.81 ± 3.46 | 72.87 ± 2.97 | 83.96 ± 9.03 | 60.42 ± 3.29 |
| | | ECGFounder | 50.00 ± 0.00 | 71.11 ± 9.13 | 43.89 ± 4.56 | 42.69 ± 4.74 | 37.72 ± 7.04 | 73.70 ± 2.42 |
| | | SL-S4Wave(Ours) | 100.00 ± 0.00 | 80.95 ± 8.52 | **84.62 ± 6.88** | **72.53 ± 8.69** | **57.62 ± 10.50** | **99.05 ± 1.90** |
| EBR

N=90 | Supervised | FCN | 100.00 ± 0.00 | 50.00 ± 0.00 | 78.57 ± 0.00 | 84.21 ± 0.00 | 72.73 ± 0.00 | 82.92 ± 0.00 |
| | | S4Wave | 65.00 ± 12.25 | 86.67 ± 6.67 | 42.67 ± 9.04 | 73.64 ± 7.96 | 87.43 ± 6.66 | 75.83 ± 4.86 |
| | Self-supervised | SimCLR | 95.00 ± 6.12 | 50.00 ± 0.00 | 69.37 ± 11.28 | 81.64 ± 3.15 | 71.64 ± 1.34 | 81.25 ± 8.54 |
| | | CMSC | 97.50 ± 5.00 | 36.67 ± 16.33 | 68.25 ± 9.01 | 79.72 ± 3.72 | 67.71 ± 5.13 | 70.21 ± 8.73 |
| | | TS-TCC | 60.00 ± 22.91 | 83.33 ± 0.00 | 42.08 ± 20.22 | 67.31 ± 17.02 | 81.00 ± 5.61 | 68.13 ± 18.70 |
| | | TF-C | 90.00 ± 12.25 | 56.67 ± 8.17 | 66.36 ± 20.95 | 80.63 ± 8.38 | 73.21 ± 5.97 | 83.33 ± 11.56 |
| | | TS2Vec | 97.50 ± 5.00 | 56.67 ± 8.17 | 76.51 ± 8.18 | 84.78 ± 2.18 | 75.19 ± 3.10 | 88.33 ± 5.37 |
| | | ECG-JEPA | 51.11 ± 9.94 | 73.34 ± 14.91 | 27.84 ± 2.66 | 60.00 ± 5.59 | 77.14 ± 12.78 | 76.30 ± 3.31 |
| | | ECGFounder | 88.89 ± 34.31 | 70.00 ± 25.39 | 69.84 ± 18.53 | 84.59 ± 34.98 | 81.59 ± 37.21 | 94.44 ± 36.77 |
| | | SL-S4Wave(Ours) | 100.00 ± 0.00 | 83.33 ± 0.00 | **92.86 ± 0.00** | **94.12 ± 0.00** | **88.89 ± 0.00** | **91.67 ± 0.00** |
| ETC

N=139 | Supervised | FCN | 94.17 ± 2.04 | 40.00 ± 20.00 | 74.47 ± 6.84 | 94.56 ± 1.68 | 94.96 ± 1.65 | 77.92 ± 18.29 |
| | | S4Wave | 90.00 ± 2.04 | 100.00 ± 0.00 | 66.56 ± 4.93 | 94.72 ± 1.14 | **100.00 ± 0.00** | 95.00 ± 1.02 |
| | Self-supervised | SimCLR | 93.33 ± 3.33 | 40.00 ± 20.00 | 72.58 ± 9.89 | 94.10 ± 2.52 | 94.89 ± 1.79 | 56.67 ± 24.80 |
| | | CMSC | 90.83 ± 1.67 | 50.00 ± 0.00 | 65.70 ± 3.90 | 93.15 ± 0.93 | 95.61 ± 0.08 | 61.25 ± 9.46 |
| | | TS-TCC | 74.17 ± 5.53 | 50.00 ± 0.00 | 37.73 ± 6.94 | 83.07 ± 3.59 | 94.65 ± 0.37 | 60.42 ± 4.37 |
| | | TF-C | 80.83 ± 6.24 | 60.00 ± 20.00 | 47.65 ± 9.34 | 87.66 ± 3.84 | 96.07 ± 2.00 | 64.17 ± 12.39 |
| | | TS2Vec | 95.00 ± 4.86 | 60.00 ± 20.00 | **80.34 ± 16.13** | **95.74 ± 3.07** | 96.59 ± 1.72 | 66.67 ± 17.13 |
| | | ECG-JEPA | 65.00 ± 5.74 | 20.00 ± 44.72 | 26.81 ± 3.46 | 72.87 ± 2.97 | 83.96 ± 9.03 | 60.42 ± 3.29 |
| | | ECGFounder | 74.17 ± 4.56 | 40.00 ± 22.36 | 37.06 ± 5.11 | 80.51 ± 3.26 | 88.24 ± 3.63 | 66.25 ± 7.13 |
| | | SL-S4Wave(Ours) | 91.67 ± 0.00 | 100.00 ± 0.00 | 70.59 ± 0.00 | 95.65 ± 0.00 | **100.00 ± 0.00** | **95.83 ± 0.00** |
| VFB

N=58 | Supervised | FCN | 100.00 ± 0.00 | 0.00 ± 0.00 | 15.39 ± 0.00 | 26.66 ± 0.00 | 15.38 ± 0.00 | 50.00 ± 0.00 |
| | | S4Wave | 100.00 ± 0.00 | 72.73 ± 14.37 | **76.92 ± 12.16** | **59.65 ± 14.09** | **43.71 ± 15.14** | 96.36 ± 8.13 |
| | Self-supervised | SimCLR | 100.00 ± 0.00 | 0.00 ± 0.00 | 15.39 ± 0.00 | 26.66 ± 0.00 | 15.38 ± 0.00 | 50.00 ± 0.00 |
| | | CMSC | 100.00 ± 0.00 | 0.00 ± 0.00 | 15.39 ± 0.00 | 26.66 ± 0.00 | 15.38 ± 0.00 | 50.00 ± 0.00 |
| | | TS-TCC | 20.00 ± 24.49 | 32.73 ± 9.27 | 21.29 ± 8.07 | 8.00 ± 9.80 | 5.00 ± 6.12 | 28.64 ± 16.98 |
| | | TF-C | 80.00 ± 24.49 | 49.09 ± 15.85 | 50.23 ± 19.78 | 36.70 ± 15.18 | 24.11 ± 11.05 | 68.18 ± 17.49 |
| | | TS2Vec | 100.00 ± 0.00 | 23.64 ± 7.27 | 35.38 ± 6.15 | 32.40 ± 2.22 | 19.35 ± 1.60 | 70.00 ± 13.67 |
| | | ECG-JEPA | 0.00 ± 0.00 | 68.89 ± 14.18 | 44.28 ± 9.31 | 0.00 ± 0.00 | 0.00 ± 0.00 | 50.00 ± 0.00 |
| | | ECGFounder | 0.00 ± 0.00 | 75.56 ± 12.67 | 48.57 ± 7.9 | 0.00 ± 0.00 | 0.00 ± 0.00 | 50.00 ± 0.00 |
| | | SL-S4Wave(Ours) | 100.00 ± 0.00 | 72.73 ± 12.86 | **76.92 ± 10.88** | 59.17 ± 12.88 | 43.05 ± 14.10 | **97.27 ± 4.07** |
| VTA

N=343 | Supervised | FCN | 97.65 ± 2.88 | 28.52 ± 4.16 | 44.17 ± 4.27 | 46.03 ± 2.09 | 30.12 ± 1.56 | 68.69 ± 4.47 |
| | | S4Wave | 87.06 ± 4.92 | 87.41 ± 5.62 | 77.74 ± 2.75 | 76.97 ± 4.43 | 69.70 ± 9.03 | **92.11 ± 2.84** |
| | Self-supervised | SimCLR | 90.59 ± 4.71 | 25.56 ± 5.90 | 37.68 ± 2.08 | 42.44 ± 0.76 | 27.74 ± 0.76 | 56.60 ± 4.73 |
| | | CMSC | 89.41 ± 2.35 | 28.89 ± 2.77 | 39.42 ± 2.48 | 43.08 ± 1.33 | 28.38 ± 0.99 | 59.13 ± 3.03 |
| | | TS-TCC | 80.00 ± 6.00 | 60.74 ± 4.60 | 54.97 ± 3.64 | 52.54 ± 3.03 | 39.21 ± 2.72 | 75.19 ± 1.15 |
| | | TF-C | 67.06 ± 7.06 | 49.63 ± 16.62 | 40.60 ± 6.34 | 41.68 ± 3.84 | 31.03 ± 5.81 | 63.14 ± 3.97 |
| | | TS2Vec | 88.24 ± 6.44 | 36.30 ± 5.93 | 43.82 ± 2.48 | 45.17 ± 1.64 | 30.41 ± 1.22 | 69.52 ± 4.04 |
| | | ECG-JEPA | 57.28 ± 6.10 | 71.15 ± 4.61 | 44.52 ± 2.56 | 50.75 ± 3.01 | 45.83 ± 3.17 | 74.27 ± 0.42 |
| | | ECGFounder | 67.27 ± 3.80 | 71.92 ± 3.75 | 50.84 ± 3.01 | 57.62 ± 2.99 | 50.49 ± 3.48 | 77.62 ± 2.30 |
| | | SL-S4Wave(Ours) | 94.12 ± 3.72 | 87.78 ± 1.48 | **84.67 ± 4.17** | **80.82 ± 2.93** | **70.85 ± 2.91** | 90.95 ± 2.26 |

Table 4: **Transferability to other types of arrhythmias: Classification results on the PhysioNet Challenge 2015 dataset.** We pre-trained on the unlabeled dataset from VTaC and fine-tuned on the labeled data from the Challenge 2015 data. We report the test set performance for four distinct classes of life-threatening arrhythmias from the Challenge 2015 dataset, including Asystole (ASY), Extreme Bradycardia (EBR), Extreme Tachycardia (ETC), Ventricular Flutter/Fibrillation (VFB), and Ventricular Tachycardia (VTA). Performance was reported as the mean and standard deviation over five random seeds. Despite the limited size of the target dataset, SL-S4Wave demonstrated strong cross-domain transferability across different arrhythmia classes. The result in this table uses the S4 d state=64 setting.

We designed experiments to explore the performance of the S4Wave model under different SSL strategies. We tested the performance of three different self-supervised pre-training frameworks, SimCLR, TS-TCC, and TS2Vec, on the VTaC dataset. Table 7 illustrates the results. Among the three SSL frameworks, TS-TCC achieves the highest TPR (94.16%), while TS2Vec leads in AUC (95.53%), and SimCLR shows competitive

| Datasets | Method | | TPR | TNR | Score | F1 | PPV | AUC |
|---|---|---|---|---|---|---|---|---|
| ASY

N=944 | Supervised | FCN | 88.00 ± 0.00 | 56.30 ± 4.00 | 56.54 ± 2.40 | 17.76 ± 2.20 | 9.88 ± 0.88 | 83.00 ± 0.52 |
| | | S4Wave | 80.00 ± 0.00 | 50.43 ± 17.89 | 49.12 ± 28.63 | 16.81 ± 26.29 | 9.67 ± 4.50 | 65.22 ± 2.98 |
| | Self-supervised | SimCLR | 100.00 ± 0.00 | 31.52 ± 0.00 | 35.05 ± 5.16 | 13.76 ± 4.89 | 7.39 ± 0.92 | 86.37 ± 0.53 |
| | | CMSC | 100.00 ± 0.00 | 29.13 ± 0.00 | 32.78 ± 2.85 | 13.32 ± 2.85 | 7.13 ± 0.48 | 84.97 ± 0.27 |
| | | TS-TCC | 56.00 ± 0.00 | 33.33 ± 19.60 | 27.47 ± 20.43 | 25.13 ± 12.96 | **16.59 ± 3.24** | 42.57 ± 1.70 |
| | | TF-C | 98.00 ± 0.00 | 39.89 ± 4.00 | 42.71 ± 3.05 | 15.05 ± 2.83 | 8.15 ± 0.72 | 83.21 ± 0.41 |
| | | TS2Vec | 98.00 ± 0.00 | 1.09 ± 4.00 | 6.05 ± 1.14 | 9.71 ± 0.89 | 5.11 ± 0.29 | 41.73 ± 0.15 |
| | | ECG-JEPA | 15.17 ± 15.01 | 30.08 ± 28.04 | 25.20 ± 23.43 | 4.14 ± 3.70 | 2.83 ± 2.55 | 10.77 ± 6.12 |
| | | ECGFounder | 70.00 ± 9.35 | 65.22 ± 7.69 | 61.75 ± 8.18 | 17.91 ± 5.30 | 10.31 ± 3.26 | 78.12 ± 8.63 |
| | | SL-S4Wave(Ours) | 94.00 ± 0.00 | 74.13 ± 4.90 | **74.25 ± 1.56** | **28.11 ± 1.54** | 16.53 ± 1.95 | **92.17 ± 1.24** |
| EBR

N=958 | Supervised | FCN | 90.83 ± 3.12 | 49.07 ± 4.41 | 54.46 ± 4.66 | 51.99 ± 2.98 | 36.45 ± 2.55 | 70.14 ± 1.81 |
| | | S4Wave | 89.17 ± 5.65 | 44.80 ± 31.13 | 49.39 ± 17.96 | 52.42 ± 10.99 | 38.75 ± 12.05 | 66.98 ± 12.76 |
| | Self-supervised | SimCLR | 88.33 ± 6.67 | 34.40 ± 5.23 | 42.75 ± 3.31 | 44.90 ± 2.04 | 30.14 ± 1.39 | 70.14 ± 1.81 |
| | | CMSC | 90.00 ± 2.04 | 33.87 ± 2.99 | 43.30 ± 2.36 | 45.40 ± 1.31 | 30.36 ± 1.06 | 71.37 ± 2.75 |
| | | TS-TCC | 92.50 ± 15.00 | 60.00 ± 22.61 | **71.98 ± 18.70** | **82.89 ± 7.88** | **77.11 ± 8.99** | **78.75 ± 5.99** |
| | | TF-C | 94.17 ± 3.33 | 33.87 ± 5.24 | 45.90 ± 3.42 | 47.04 ± 1.75 | 31.37 ± 1.50 | 75.64 ± 3.18 |
| | | TS2Vec | 100.00 ± 0.00 | 2.67 ± 3.48 | 26.26 ± 2.63 | 39.69 ± 0.88 | 24.76 ± 0.69 | 44.91 ± 9.54 |
| | | ECG-JEPA | 64.17 ± 27.10 | 39.20 ± 26.47 | 33.18 ± 6.43 | 34.86 ± 8.26 | 25.15 ± 3.79 | 54.79 ± 8.28 |
| | | ECGFounder | 63.33 ± 7.98 | 61.33 ± 7.69 | 45.75 ± 5.60 | 44.70 ± 5.74 | 34.76 ± 5.45 | 71.94 ± 9.13 |
| | | SL-S4Wave(Ours) | 88.33 ± 4.08 | 60.00 ± 6.69 | 60.19 ± 5.52 | 56.61 ± 4.15 | 41.79 ± 4.16 | 74.17 ± 4.06 |
| ETC

N=2873 | Supervised | FCN | 94.27 ± 1.92 | 46.54 ± 2.33 | 64.23 ± 3.00 | 78.17 ± 0.82 | 66.79 ± 0.84 | 83.26 ± 0.84 |
| | | S4Wave | 91.60 ± 6.55 | 36.26 ± 23.18 | 55.86 ± 3.35 | 74.32 ± 3.09 | 63.33 ± 6.86 | 63.93 ± 8.61 |
| | Self-supervised | SimCLR | 91.26 ± 2.44 | 29.42 ± 5.65 | 52.64 ± 1.95 | 72.10 ± 0.74 | 59.64 ± 1.45 | 67.31 ± 2.64 |
| | | CMSC | 91.47 ± 1.71 | 28.09 ± 3.32 | 52.38 ± 1.34 | 71.87 ± 0.39 | 59.21 ± 0.76 | 66.13 ± 1.07 |
| | | TS-TCC | 82.50 ± 7.64 | 50.00 ± 0.00 | 50.95 ± 13.78 | **88.21 ± 4.52** | **95.16 ± 0.42** | 68.96 ± 9.07 |
| | | TF-C | 93.31 ± 0.98 | 26.85 ± 5.29 | 54.52 ± 2.65 | 72.50 ± 1.39 | 59.30 ± 1.73 | 65.54 ± 1.99 |
| | | TS2Vec | 99.59 ± 0.33 | 0.54 ± 0.91 | 52.85 ± 0.50 | 69.44 ± 0.18 | 53.31 ± 0.20 | 55.23 ± 1.75 |
| | | ECG-JEPA | 77.00 ± 22.15 | 35.33 ± 27.73 | 41.13 ± 10.21 | 64.72 ± 8.38 | 58.85 ± 4.87 | 60.34 ± 1.90 |
| | | ECGFounder | 80.61 ± 4.98 | 54.55 ± 10.15 | 48.78 ± 7.11 | 73.17 ± 4.21 | 67.14 ± 5.14 | 76.59 ± 6.73 |
| | | SL-S4Wave(Ours) | 91.95 ± 2.05 | 56.89 ± 3.45 | **64.61 ± 3.46** | 80.04 ± 1.49 | 70.89 ± 1.70 | **84.27 ± 0.95** |
| VFB

N=467 | Supervised | FCN | 97.55 ± 1.64 | 51.84 ± 2.29 | 74.33 ± 2.94 | 84.06 ± 0.52 | 73.87 ± 0.70 | 90.69 ± 1.78 |
| | | S4Wave | 95.66 ± 1.85 | 46.58 ± 23.84 | 68.43 ± 9.87 | 82.20 ± 6.06 | 72.53 ± 8.97 | 71.12 ± 12.07 |
| | Self-supervised | SimCLR | 100.00 ± 0.00 | 18.42 ± 7.11 | 65.93 ± 2.97 | 77.40 ± 1.53 | 63.16 ± 2.05 | 90.83 ± 2.17 |
| | | CMSC | 100.00 ± 0.00 | 13.68 ± 4.29 | 63.96 ± 1.79 | 76.38 ± 0.91 | 61.79 ± 1.19 | 89.24 ± 3.08 |
| | | TS-TCC | 50.00 ± 0.00 | 7.27 ± 8.91 | 10.59 ± 5.76 | 15.24 ± 1.17 | 9.00 ± 0.82 | 31.82 ± 8.38 |
| | | TF-C | 95.47 ± 0.71 | 33.16 ± 4.51 | 62.85 ± 2.41 | 78.46 ± 1.20 | 66.61 ± 1.56 | 86.51 ± 1.66 |
| | | TS2Vec | 98.87 ± 1.83 | 1.32 ± 2.63 | 56.73 ± 2.24 | 73.33 ± 0.38 | 58.29 ± 0.23 | 58.26 ± 3.23 |
| | | ECG-JEPA | 75.47 ± 31.81 | 43.16 ± 28.53 | 45.70 ± 17.00 | 66.26 ± 21.14 | 64.73 ± 5.76 | 65.83 ± 6.12 |
| | | ECGFounder | 90.38 ± 5.71 | 55.53 ± 7.11 | 62.84 ± 10.60 | 81.30 ± 4.16 | 73.95 ± 3.56 | 88.93 ± 4.63 |
| | | SL-S4Wave(Ours) | 95.66 ± 0.46 | 70.79 ± 4.28 | **77.47 ± 2.50** | **88.35 ± 1.44** | **82.09 ± 2.20** | **96.53 ± 0.39** |
| VTA

N=2768 | Supervised | FCN | 97.11 ± 0.57 | 63.33 ± 3.78 | 82.77 ± 1.47 | 94.22 ± 0.33 | 91.50 ± 0.76 | 91.15 ± 0.62 |
| | | S4Wave | 96.41 ± 2.20 | 49.82 ± 33.39 | 78.28 ± 2.59 | 92.49 ± 2.60 | 89.24 ± 6.33 | 73.12 ± 15.70 |
| | Self-supervised | SimCLR | 98.19 ± 1.21 | 36.67 ± 5.86 | 81.41 ± 3.04 | 91.86 ± 0.36 | 86.32 ± 0.99 | 86.59 ± 1.67 |
| | | CMSC | 98.70 ± 0.71 | 34.74 ± 4.14 | 82.67 ± 1.73 | 91.92 ± 0.25 | 86.01 ± 0.69 | 85.02 ± 1.71 |
| | | TS-TCC | 89.41 ± 6.86 | 45.19 ± 17.08 | 50.62 ± 10.76 | 50.08 ± 6.31 | 35.26 ± 6.29 | 68.81 ± 8.11 |
| | | TF-C | 95.81 ± 0.32 | 41.58 ± 4.98 | 75.02 ± 1.46 | 91.17 ± 0.61 | 86.96 ± 0.97 | 85.61 ± 2.00 |
| | | TS2Vec | 99.05 ± 0.77 | 0.70 ± 0.66 | 77.32 ± 2.30 | 88.63 ± 0.32 | 80.20 ± 0.05 | 47.79 ± 2.56 |
| | | ECG-JEPA | 70.76 ± 31.29 | 41.75 ± 31.19 | 42.71 ± 23.39 | 72.42 ± 24.83 | 83.52 ± 2.39 | 62.30 ± 6.12 |
| | | ECGFounder | 91.40 ± 2.15 | 70.88 ± 4.72 | 68.67 ± 4.81 | 92.05 ± 1.09 | 92.74 ± 1.04 | 89.93 ± 1.12 |
| | | SL-S4Wave(Ours) | 96.59 ± 0.44 | 81.05 ± 1.63 | **84.30 ± 1.20** | **95.99 ± 0.15** | **95.40 ± 0.36** | **95.04 ± 0.51** |

Table 5: **Transferability to other types of arrhythmias: classification results on the MIMIC II Arrhythmia dataset (Total N=8,010).** We pre-trained on the unlabeled VTaC dataset and fine-tuned on five different disease-specific sub-datasets. ASY, EBR, ETC, VFB, and VTA represent Asystole, Extreme Bradycardia, Extreme Tachycardia, Ventricular Tachycardia, and Ventricular Flutter/Fibrillation, respectively.

performance in TNR (86.38%) and F1 (80.78%). Overall, all three SSL-pretrained variants outperform the baseline S4Wave across most metrics, demonstrating that self-supervised pre-training provides a beneficial initialization for the model. However, these generic time-series SSL frameworks are not specifically designed for physiological waveform data, which may limit their ability to capture domain-specific temporal patterns. In contrast, SL-S4Wave, which incorporates our proposed self-supervised learning strategy tailored for physiological signals, achieves the best performance across all metrics, with a Score of 83.25 ± 0.44, F1 of 82.89 ± 0.97, PPV of 73.88 ± 2.41, and AUC of 96.01 ± 0.29. Notably, SL-S4Wave also exhibits the smallest

| Dataset | Method | TPR | TNR | Score | F1 | PPV | AUC |
|---|---|---|---|---|---|---|---|
| VTaC | FCN w/o Pre | 91.83 ± 1.31 | 86.96 ± 2.60 | 80.83 ± 1.65 | 81.79 ± 2.15 | 73.82 ± 3.70 | 94.93 ± 0.38 |
| | FCN | 91.82±2.50 | 84.87±1.58 | 79.52±2.26 | 79.88±1.23 | 70.74±1.87 | 94.47±0.79 |
| | ResNet w/o Pre | 55.33±6.68 | 67.94±6.32 | 42.79±3.51 | 46.92±4.34 | 41.11±5.25 | 68.90±3.54 |
| | ResNet | 65.26±0.88 | 79.30±1.39 | 53.99±1.26 | 60.06±1.50 | 55.64±1.93 | 79.76±1.68 |
| | Transformer w/o Pre | 94.82 ± 1.87 | 18.64 ± 3.73 | 51.61 ± 2.13 | 68.88 ± 0.85 | 54.10 ± 0.94 | 64.76 ± 1.60 |
| | Transformer | 83.36 ± 2.39 | 67.36 ± 4.51 | 60.45 ± 0.88 | 62.84 ± 1.54 | 50.55 ± 2.93 | 84.39 ± 0.86 |
| | S4Wave | 91.24 ± 1.33 | 83.30 ± 1.44 | 77.84 ± 1.14 | 78.27 ± 0.97 | 68.36 ± 1.52 | 94.20 ± 0.50 |
| | SL-S4Wave | 93.98 ± 1.55 | 86.96 ± 1.99 | **83.25 ± 0.44** | **82.89 ± 0.97** | **73.88 ± 2.41** | **96.01 ± 0.29** |
| MIMIC | FCN w/o Pre | 92.41±1.72 | 52.54±4.00 | 63.01±2.98 | 77.25±1.59 | 66.45±2.08 | 85.87±1.00 |
| | FCN | 77.88±1.82 | 83.47±1.10 | 66.34±0.76 | 69.74±0.12 | 63.19±1.07 | 87.32±0.03 |
| | ResNet w/o Pre | 94.11±0.28 | 25.16±1.21 | 53.49±0.15 | 70.19±0.18 | 55.97±0.33 | 74.80±0.39 |
| | ResNet | 93.40±1.58 | 34.05±7.32 | 56.39±1.35 | 72.27±1.34 | 59.00±2.32 | 78.28±1.33 |
| | Transformer w/o Pre | 67.15 ±1.74 | 66.44 ±4.95 | 49.13 ±1.24 | 53.12 ±0.62 | 44.75 ±1.04 | 72.76 ±2.29 |
| | Transformer | 97.09 ± 1.74 | 11.83 ± 4.95 | 51.68 ± 1.24 | 68.30 ± 0.62 | 52.70 ± 1.04 | 62.54 ± 2.29 |
| | S4Wave | 90.92 ± 3.41 | 46.88 ± 8.27 | 58.44 ± 2.04 | 74.71 ± 1.08 | 63.39 ± 2.88 | 81.57 ± 1.00 |
| | SL-S4Wave | 94.37 ± 0.77 | 60.21 ± 2.02 | **69.57 ± 1.18** | **80.83 ± 0.59** | **70.86 ± 0.90** | **88.56 ± 0.36** |

Table 6: Effect of Encoder Architecture and Pre-training in SL-S4Wave: Evaluation of pre-training effectiveness using different baseline models. The ResNet we are using here is ResNet18. Our results demonstrate that S4Wave is an effective encoder architecture for SL-S4Wave, capturing temporal dependencies from long, multi-channel physiological signals more effectively than convolutional or attention-based encoders, such as FCN, ResNet and Transformer. All models use 10s segment of waveforms as input for classification.

Table 7: Performance of S4Wave under different SSL strategies on VTaC dataset.

| Method | TPR | TNR | Score | F1 | PPV | AUC |
|---|---|---|---|---|---|---|
| S4Wave | 91.24 ± 1.33 | 83.30 ± 1.44 | 77.84 ± 1.14 | 78.27 ± 0.97 | 68.36 ± 1.52 | 94.20 ± 0.50 |
| SimCLR | 90.95 ± 1.90 | 86.38 ± 2.49 | 79.51 ± 1.09 | 80.78 ± 1.48 | 72.36 ± 3.34 | 95.12 ± 0.34 |
| TS-TCC | 94.16 ± 1.63 | 84.12 ± 3.62 | 81.55 ± 0.83 | 80.50 ± 2.37 | 70.09 ± 4.48 | 95.13 ± 0.43 |
| TS2Vec | 93.43 ± 2.19 | 83.88 ± 1.94 | 80.61 ± 1.62 | 79.86 ± 1.04 | 69.42 ± 2.16 | 95.53 ± 0.59 |
| SL-S4Wave | **93.98 ± 1.55** | **86.96 ± 1.99** | **83.25 ± 0.44** | **82.89 ± 0.97** | **73.88 ± 2.41** | **96.01 ± 0.29** |

standard deviation on Score (0.44) and AUC (0.29) among all methods, indicating superior training stability. These results suggest that a domain-aware self-supervised strategy is more effective than general-purpose SSL frameworks for waveform-based clinical tasks.

## B.4 Performance with Different Block Number

Since our model consists of several S4Wave blocks stacked together, to demonstrate the impact of different numbers of blocks on the model performance, we conducted an ablation study to explore the effect of multiple blocks on model performance. Due to the purpose of the experiment was solely to explore the effects brought about by the model structure, we did not pre-train the different block variants of the model on the VTaC unlabelled dataset; instead, we directly fine-tuned them on the VTaC dataset. In Fig. 5, we illustrate the performance with block number= $[1, 2, 3, 4, 5, 6]$.

We can see that as the number of blocks increases from 1 to 3 layers, both the AUC and Score show a significant improvement. However, when using larger numbers of blocks, although the model's performance continues to increase, it is accompanied by larger standard deviations. In other words, the number of blocks generally follows the principle of diminishing marginal utility, with the utility being maximized at three blocks. While there is still some improvement beyond three layers, the overall increase is marginal. This may be because the model has entered a state of overfitting when the number of blocks exceeds three. We did not attempt larger blocks due to computational resource constraints. However, we can speculate that larger blocks might still offer some performance improvements, but the potential for further increases would likely be limited.

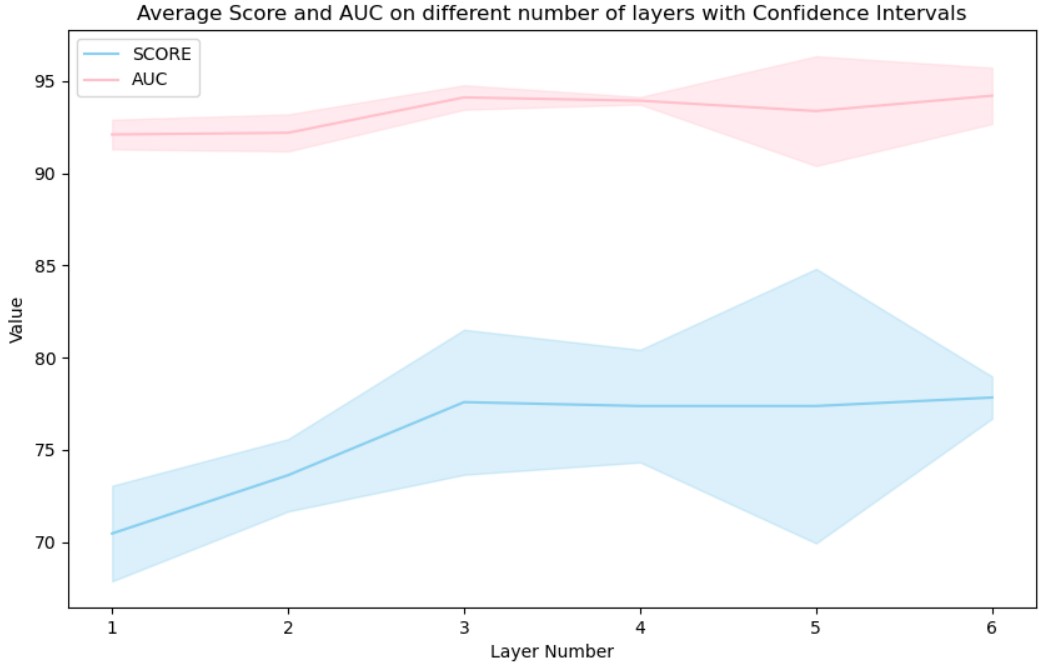

Figure 5: The results of SL-S4Wave on the VTaC dataset with different numbers of blocks.

### B.5 Parameter Sensitivity

In the pre-training, we used $\lambda$ as a parameter to adjust the roles of the two parts of the loss. In Table. 8, we tested the pre-training results on the VTaC unlabelled dataset with $\lambda$ values of 0.01, 0.1, 1, 10, and 100, and then fine-tuned on the VTaC dataset.

As Table. 8 shows, The model appears to be relatively insensitive to the $\lambda$ parameter. When the Context Consistency Loss has a greater impact ($\lambda = 10, 100$), there is a slight but not significant decrease in performance. When the Noise-resilient Loss plays a larger role ($\lambda = 0.1, 0.01$), there is virtually no change in the model's performance. Overall, adjusting the $\lambda$ parameter has almost no effect on the fine-tuning performance of the model, and those performance variations are likely due more to initialized parameters.

| $\lambda$ | TPR | TNR | Score | F1 | PPV | AUC |
|---|---|---|---|---|---|---|
| 0.01 | **93.43** | 84.49 | 80.99 | 80.37 | 70.15 | 94.40 |
| 0.1 | **93.43** | 84.49 | 80.99 | 80.37 | 70.15 | 94.40 |
| 1 | **93.43** | **84.99** | **81.32** | 80.83 | 70.89 | **95.25** |
| 10 | 92.52 | 86.09 | 81.03 | **81.31** | **72.14** | 94.62 |
| 100 | 91.97 | 85.80 | 80.25 | 80.77 | 71.63 | 94.63 |

Table 8: Parameter sensitivity experiment on VTaC dataset. We reported the average of 5 experiments. The best result is highlighted in bold. The experiment is conducted at d=64.

### B.6 Performance Comparison of Time Lengths

Table 9 and Table 10 record the complete data of Figure 3 in the section "Performance of SL-S4Wave on Longer Sequences." Each method in the table has undergone hyperparameter tuning. We performed a grid

search from lr: 1e-3, 1e-4, 1e-5, batch size: 32, 64, 128 and selected the AUC metric in best validation set on five different random seeds each time. The mean and standard deviation are reported.

| Method | Time | TPR | TNR | Score | F1 | PPV | AUC |
|---|---|---|---|---|---|---|---|
| SL-S4Wave | 10s | 93.98 ± 1.55 | 86.96 ± 1.99 | 83.25 ± 0.44 | 82.89 ± 0.97 | 73.78 ± 2.41 | 96.01 ± 0.29 |
| | 20s | 93.72 ± 1.76 | 89.39 ± 0.97 | 84.61 ± 1.55 | 85.03 ± 0.54 | 77.85 ± 1.34 | 96.41 ± 0.47 |
| | 30s | 96.35 ± 0.73 | 86.96 ± 1.49 | **86.06 ± 1.51** | **84.09 ± 1.53** | **74.63 ± 2.25** | **96.70 ± 0.49** |
| S4Wave | 10s | 91.24 ± 1.33 | 83.30 ± 1.44 | 77.84 ± 1.14 | 82.59 ± 0.97 | 68.36 ± 1.52 | 94.20 ± 0.50 |
| | 20s | 94.01 ± 1.19 | 87.83 ± 1.30 | 83.90 ± 0.89 | 78.27 ± 1.05 | **74.99 ± 2.01** | 96.05 ± 0.67 |
| | 30s | 96.17 ± 2.44 | 86.09 ± 0.85 | **85.24 ± 0.78** | **83.23 ± 2.03** | 73.02 ± 1.78 | **96.45 ± 0.61** |
| FCN | 10s | 91.83 ± 1.31 | 86.96 ± 2.60 | **80.83 ± 1.65** | **81.79 ± 2.15** | 73.82 ± 3.70 | **94.93 ± 0.38** |
| | 20s | 85.40 ± 4.63 | 84.93 ± 3.59 | 72.95 ± 3.33 | 76.47 ± 1.93 | 68.82 ± 2.89 | 92.06 ± 0.72 |
| | 30s | 87.10 ± 1.32 | 76.81 ± 5.25 | 69.66 ± 0.98 | 70.98 ± 0.82 | 60.16 ± 1.79 | 89.71 ± 0.84 |
| CNN | 10s | 95.04 ± 0.85 | 74.32 ± 0.70 | **75.92 ± 0.65** | **73.26 ± 0.40** | **59.46 ± 0.60** | **93.45 ± 0.37** |
| | 20s | 97.08 ± 2.08 | 70.20 ± 3.92 | 75.33 ± 1.14 | 71.47 ± 1.83 | 56.41 ± 3.01 | 93.38 ± 0.22 |
| | 30s | 95.04 ± 0.80 | 70.20 ± 4.50 | 73.14 ± 2.58 | 70.38 ± 2.73 | 55.67 ± 3.58 | 92.46 ± 0.56 |
| SimCLR | 10s | 91.83 ± 0.40 | 86.96 ± 2.18 | **80.10 ± 1.53** | 77.14 ± 1.68 | 64.23 ± 2.31 | **93.73 ± 0.07** |
| | 20s | 85.11 ± 0.85 | 86.26 ± 1.57 | 73.52 ± 0.91 | **77.51 ± 0.98** | **70.90 ± 1.49** | 92.98 ± 0.17 |
| | 30s | 89.93 ± 1.42 | 81.45 ± 0.73 | 75.26 ± 1.39 | 76.00 ± 0.78 | 65.47 ± 0.84 | 91.48 ± 0.34 |

Table 9: Results using different sequence lengths with SL-S4Wave and baseline model in the VTaC dataset.

| Method | Time | TPR | TNR | Score | F1 | PPV | AUC |
|---|---|---|---|---|---|---|---|
| SL-S4Wave | 10s | 94.37 ± 0.77 | 60.21 ± 2.02 | 69.57 ± 1.18 | 80.83 ± 0.59 | 70.86 ± 0.90 | 88.56 ± 0.36 |
| | 20s | 94.68 ± 1.50 | 60.86 ± 6.97 | 70.34 ± 1.65 | **81.18 ± 1.77** | **71.18 ± 3.33** | 90.11 ± 0.94 |
| | 30s | 96.10 ± 1.23 | 57.06 ± 6.48 | **71.09 ± 1.20** | 80.61 ± 1.64 | 69.51 ± 3.07 | **91.14 ± 0.57** |
| S4Wave | 10s | 90.92 ± 3.04 | 46.88 ± 8.27 | 58.44 ± 2.41 | 74.71 ± 1.08 | 63.39 ± 2.88 | 81.57 ± 1.00 |
| | 20s | 92.48 ± 2.14 | 58.49 ± 6.44 | 65.70 ± 2.15 | 79.23 ± 1.49 | 69.41 ± 2.81 | 89.07 ± 1.04 |
| | 30s | 93.55 ± 2.48 | 59.14 ± 4.45 | **67.77 ± 2.80** | **79.97 ± 0.95** | **69.91 ± 1.81** | **89.74 ± 0.62** |
| FCN | 10s | 92.41 ± 1.72 | 52.54 ± 4.00 | **63.01 ± 2.98** | 77.25 ± 1.59 | 66.45 ± 2.08 | 85.87 ± 1.00 |
| | 20s | 90.43 ± 1.23 | 51.04 ± 5.64 | 59.40 ± 1.43 | 75.75 ± 1.32 | 65.23 ± 2.37 | 84.26 ± 2.13 |
| | 30s | 88.58 ± 3.21 | 59.35 ± 4.95 | 60.35 ± 2.48 | **77.43 ± 0.41** | **68.89 ± 1.88** | **85.95 ± 1.10** |
| CNN | 10s | 95.46 ± 3.53 | 15.27 ± 10.88 | 51.02 ± 0.78 | **68.37 ± 1.04** | 53.25 ± 2.34 | 70.94 ± 2.24 |
| | 20s | 96.03 ± 3.81 | 10.97 ± 8.83 | 49.75 ± 1.69 | 67.64 ± 0.63 | 52.24 ± 1.66 | 70.04 ± 3.28 |
| | 30s | 99.86 ± 0.28 | 0.14 ± 0.29 | 50.13 ± 0.28 | 66.87 ± 0.06 | 50.18 ± 0.00 | 50.52 ± 3.87 |
| SimCLR | 10s | 97.38 ± 0.40 | 17.56 ± 2.18 | **54.76 ± 3.42** | **69.82 ± 1.68** | **54.33 ± 2.31** | **71.42 ± 4.41** |
| | 20s | 95.46 ± 3.81 | 15.27 ± 8.83 | 51.02 ± 1.69 | 68.37 ± 0.63 | 53.25 ± 1.66 | 70.94 ± 3.28 |
| | 30s | 100.00 ± 0.00 | 0.00 ± 0.00 | 50.27 ± 0.00 | 66.90 ± 0.00 | 50.18 ± 0.00 | 49.85 ± 0.15 |

Table 10: Results using different sequence lengths with SL-S4Wave and baseline model in the MIMIC II-VT dataset.

## B.7 Efficiency Analysis

In order to further analyze the computational efficiency of the S4Wave model on long sequences, we report the FLOPS, GPU memory, and training time per iteration for both the S4Wave model and Transformer on 10s, 20s, and 30s of the VTaC dataset, as well as their parameter numbers. We utilize the Transformer mentioned in Appendix B.3 because the number of parameters for the Transformer using these hyperparameters is not much different from that of the S4Wave we used in the previous table. As shown in Table 11, both models exhibit increasing FLOPs and GPU memory consumption as the input sequence length grows. However, the two models differ markedly in their scaling behavior. For S4Wave, GPU memory usage scales approximately linearly with sequence length, increasing from 2.52G at 10s to 7.41G at 30s (a 2.94× increase), which is consistent with the linear complexity of the SSM-based architecture with respect to sequence length. In contrast, the Transformer's GPU memory grows from 4.62G to 13.51G over the same range (a 2.92× increase in absolute terms), but starts from a substantially higher baseline—consuming 1.83× more memory than S4Wave at 10s, and 1.82× more at 30s. This persistent memory gap highlights the practical advantage of S4Wave in memory-constrained settings, particularly when processing longer physiological recordings. Regarding training speed, S4Wave processes sequences at 1.6 it/s for 10s inputs and maintains a relatively stable throughput of 1.0 it/s at 30s. The Transformer achieves a higher raw speed of 3.39 it/s on short 10s sequences, likely due to its highly optimized parallel attention implementation, but degrades sharply

Table 11: Efficiency Analysis on S4Wave and Transformer.

| Method | Length | FLOPs | GPU Memory | Training Speed | Parameters |
|--------|--------|-------|-----------|----------------|------------|
| S4Wave | 10s | 34.4G | 2.52G | 1.6 it/s | 18.1M |
|        | 20s | 69.0G | 4.99G | 1.1 it/s | 18.1M |
|        | 30s | 103.4G | 7.41G | 1.0 it/s | 18.2M |
| Transformer | 10s | 31.5G | 4.62G | 3.39 it/s | 18.9M |
|             | 20s | 62.9G | 9.07G | 1.40 it/s | 18.9M |
|             | 30s | 96.4G | 13.51G | 0.75 it/s | 18.9M |

to 0.75 it/s at 30s—falling below S4Wave's throughput. This reversal indicates that while the Transformer benefits from parallelism on short sequences, its quadratic attention mechanism imposes a heavier computational burden as sequence length increases, making S4Wave a more scalable choice for long-sequence clinical waveform analysis.

## C  The SL-S4Wave with Different Kernel Sizes

According to Li et al. (2022) paper, the effectiveness of SGConv primarily stems from two features: **Parameterization Efficiency** and **Weight Decay**.

**Parameterization Efficiency:** Unlike local convolutions, SGConv is a global convolution algorithm, meaning it directly employs a convolution kernel whose length matches that of the sequence. However, naively parameterizing the convolution kernel as in classical local convolution is problematic for long sequences. SGConv constructs its global convolution kernel by concatenating smaller convolution kernels of varying lengths; each larger sub-kernel doubles the size of the previous one, while the number of parameters at each scale remains constant. This design ensures that the number of parameters depends logarithmically on the input length. Its number of kernels $N$ is given by the following equation:

$$N = log_2(\frac{L}{d}) + 1 \tag{17}$$

where $L$ is the sequence length, $d$ is the first kernel size.

**Weight Decay:** The decay of kernel weights is crucial for the model to capture long-sequence features. Specifically, the magnitude of the values in the convolution kernel should decay so that more important parts are assigned to the nearer time intervals. This allows the model to focus more on local features and reduces disturbances caused by long sequences. The initialization of a kernel parameter in SGConv is:

$$k_i = \alpha^i \text{Upsample}_{2^{max[i-1,0]}d}(w_i) \tag{18}$$

where $\alpha$ is the decay coefficient, usually chosen to be $\frac{1}{2}$. Therefore, when the sequence length is fixed, we can control the S4 kernel $K$ by first kernel size $d$. The kernel weights $k$ initialized by different $d$ are shown in Figure 6.

We recognized that the kernel size setting might have an impact on our model's results. Therefore, in the ablation experiments, we set different values of $d$ to investigate the effect of kernel size on model performance. Table. C and Table. C show the experimental results on MIMIC II dataset and VTaC dataset. Overall, smaller values of $d$ lead to better performance, which is consistent with our hypothesis. Even under the extreme condition of $d = 2$, the model outperforms those using larger kernels. Additionally, we also present the parameter counts for different $d$ values. Thanks to SGConv's efficient parameterization, smaller $d$ values correspond to fewer parameters. Thus, using SGConv not only improves model performance but also reduces computational cost. Based on the above results, we believe that in the case of globe convolutional kernels, using decay and reducing the number of parameters can effectively help the model handle complex data.

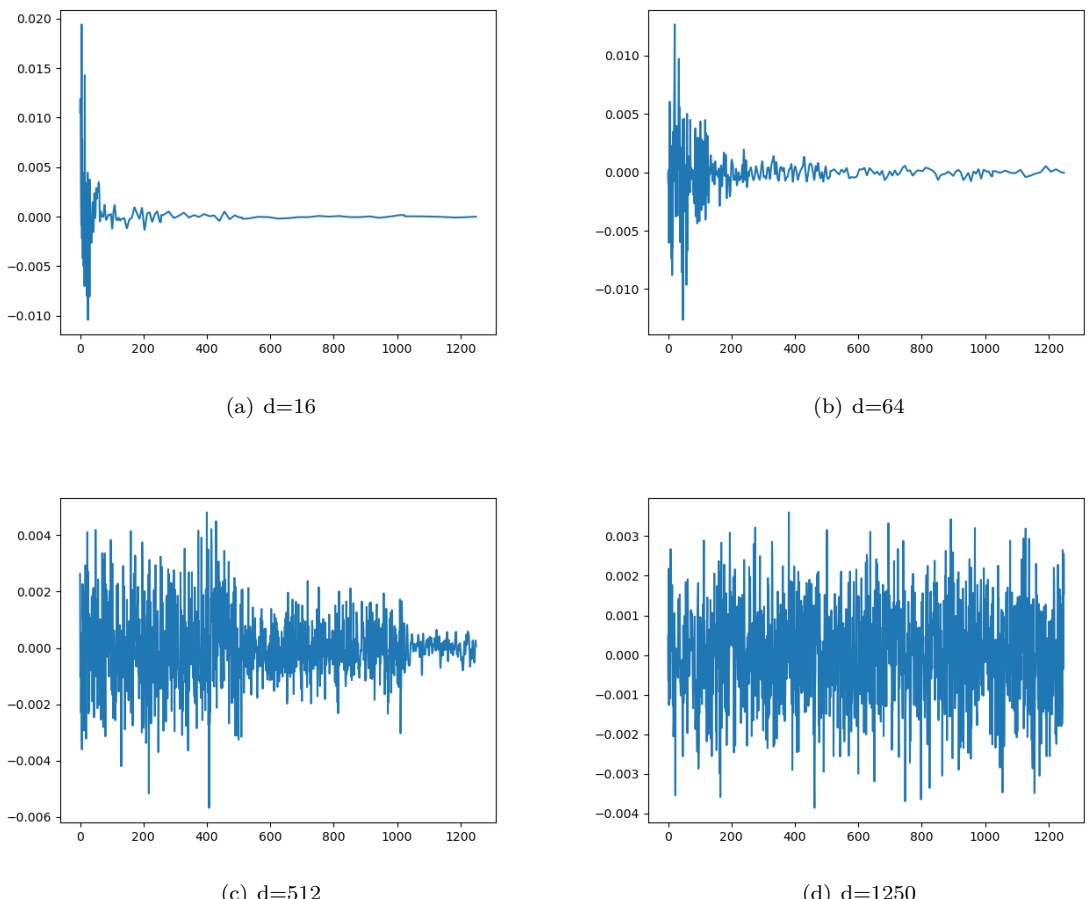

(a) d=16

(b) d=64

(c) d=512

(d) d=1250

Figure 6: Visualization of the initial weights for different kernel sizes $d$. The values in the convolution kernel exhibit a decaying behavior. In addition, as $d$ decreases, the number of parameters in the model also reduces. This efficient kernel weight design can save a significant amount of computational cost.

| d | TPR | TNR | Score | F1 | PPV | AUC | Val Score | Params |
|---|---|---|---|---|---|---|---|---|
| 2 | 89.78 | 87.36 | 80.11 | 82.13 | **74.67** | 95.42 | 82.76 | 18m |
| 16 | **93.19** | **87.38** | **82.05** | **82.59** | 74.16 | **95.66** | **82.80** | 24m |
| 64 | 91.24 | 83.30 | 77.84 | 78.27 | 68.36 | 94.20 | 82.01 | 36m |
| 128 | 88.08 | 83.48 | 74.68 | 76.73 | 67.73 | 92.86 | 79.06 | 48m |
| 512 | 85.11 | 85.86 | 73.27 | 71.77 | 70.39 | 93.16 | 76.46 | 96m |

Table 12: Results of SL-S4Wave models with different kernel sizes on the VTac dataset. We reported the average of 5 experiments. The best result is highlighted in bold.

## D   Implement details of SL-S4Wave and Baselines

**SL-S4Wave Implementation details**   Our model training consists of two stages: pre-training and fine-tuning, executed on NVIDIA-V100 GPUs. During the pre-training phase, we employed a learning rate of 0.00001, an Adam weight decay of 0.005, a batch size of 32, and trained for a maximum of 50 epochs. We also utilized 6 residual layers, with s4 lmax set to 1250, s4 $d$ state set to 2 (for Physionet Challenge 2015 dataset in table 4, s4 $d$ state set to 64), and dropout rate at 0. We use the data from 10 seconds before the

| d | TPR | TNR | Score | F1 | PPV | AUC | Val Score | Params |
|---|------|------|-------|-------|-------|-------|-----------|--------|
| 2 | 94.40 | **50.47** | **65.21** | **77.59** | **65.75** | **86.05** | **71.52** | 18m |
| 16 | 93.62 | 44.95 | 61.52 | 75.53 | 63.23 | 83.25 | 70.55 | 24m |
| 64 | **94.61** | 41.94 | 61.78 | 75.07 | 62.09 | 81.95 | 70.44 | 36m |
| 128 | 93.97 | 41.86 | 60.71 | 74.73 | 61.90 | 80.86 | 70.16 | 48m |
| 512 | 91.99 | 47.24 | 60.08 | 75.34 | 63.66 | 82.70 | 70.64 | 96m |

Table 13: Results of SL-S4Wave models with different kernel sizes on the MIMIC dataset. We reported the average of 5 experiments. The best result is highlighted in bold.

alarm and from 20 to 30 seconds before the alarm for pre-training. In the fine-tuning phase, we retained the learning rate but reduced the Adam weight decay to 0.0001 and the batch size to 16, while training for a maximum of 50 epochs. We calculate evaluation metrics from models which has the best score in validation set.

**Baseline Implementation details**  We detail the hyperparameters and model structures of the baseline methods we used in the following:

**FCN**: For the FCN model, we adopted a three-layer architecture. The hyperparameters were set as follows: learning rate at 0.0001, batch size at 64, and a maximum of 200 epochs for training. The Adam optimizer was used with a weight decay of 0.0001. The weight decay is 0.0001 and loss weight is 4 in the MIMIC II and VTaC dataset, respectively. All experiments were conducted on an NVIDIA-V100 GPU.

**SimCLR**: For SimCLR, We used the implementation by Dani Kiyasseh, which is available on CLOCS GitHub[2]. During the pre-training process, we use a learning rate of 0.0001 and weight decay of 0.0005, with a batch size of 128, and train for 50 epochs. During the fine-tuning process, we use a learning rate of 0.0001 and weight decay of 0.005, with a batch size of 32, and Loss weight set to 4. We train for 100 epochs to slightly overfit it. All experiments were conducted on an NVIDIA-V100 GPU.

**CLOCS (CMSC)**: For the CLOC model, we utilized the Contrastive Multi-segment Coding (CMSC) variant, because it performed the best in the original paper. During the pre-training process, we use a learning rate of 0.0001 and weight decay of 0.0005, with a batch size of 128, and train for 50 epochs. During the fine-tuning process, we use a learning rate of 0.0001 and weight decay of 0.005, with a batch size of 32, and Loss weight set to 4. We train for 100 epochs to slightly overfit it. All experiments were conducted on an NVIDIA-V100 GPU.

**TS-TCC**: For the TS-TCC, We used the backbone network from the original paper, which is a variant model based on Transformer. During the pre-training process, we use a learning rate of 0.00001 and weight decay of 0.0001, with a batch size of 32, and train for 50 epochs. During the fine-tuning process, we use a learning rate of 0.0001 and weight decay of 0.0001, with a batch size of 32, and Loss weight set to 4. We train for 300 epochs to slightly overfit it. All experiments were conducted on an NVIDIA-V100 GPU.

**TF-C**: For the TF-C, We used the backbone network from the original paper, which is also a variant model based on Transformer. During the pre-training process, we use a learning rate of 0.00001 and weight decay of 0.0005, with a batch size of 32, and train for 50 epochs. During the fine-tuning process, we use a learning rate of 0.000005 and weight decay of 0.0001, with a batch size of 16, and Loss weight set to 4. We train for 300 epochs to slightly overfit it. We adopt the default settings provided by the TF-C implementation for other settings. All experiments were conducted on an NVIDIA-V100 GPU.

**TS2Vec**: For the TS2Vec, During the pre-training process, we use a learning rate of 0.0001 and weight decay of 0.0005, with a batch size of 128, and train for 20 epochs. During the fine-tuning process, we use a learning rate of 0.0001 and weight decay of 0.0001, with a batch size of 16, and Loss weight set to 4. We train for 40 epochs to slightly overfit it. We adopt the default settings provided by the TS2vec implementation for other settings. All experiments were conducted on an NVIDIA-V100 GPU.

---

[2]https://github.com/danikiyasseh/CLOCS

**ECG-JEPA**: We use learning rate of 1e-4, weight decay of 1e-4, and a batch size of 64. Following the original setting, we train for 4000 iterations and apply early stopping after 1000 iterations. Since the author did not provide pre-trained weights, we used our own pre-training dataset for pre-training. During the pre-training process, we used a learning rate of 1e-4, a dropout rate of 0.1, and a weight decay of 1.0e-2. We trained for 100,000 iterations. We used the default model parameters in Github [3]. All experiments were conducted on an NVIDIA-A100 40G GPU.

**ECGFounder**: For ECGFounder, We use learning rate of 1e-5, weight decay of 1e-4, batch size of 64 and dropout rate of 0.05. Following the original setting, we train for 20000 iterations and apply early stopping after 2000 iterations. We used pre-trained weights [4]. During fine-tuning, we followed the order provided in ECGFounder ['I', 'II', 'III', 'aVR', 'aVF', 'aVL', 'V1', 'V2', 'V3', 'V4', 'V5', 'V6'] to fine-tune the model. If there are missing channels, they are filled with 0. Additionally, since our model includes CVP, Pleth, and ABP channels, we placed these two channels in the positions of V5 and V6. All experiments were conducted on an NVIDIA-A100 40G GPU.

## E  Data Preprocessing

For all datasets, each waveform recording contains six minutes of multi-channel physiological waveforms, with the arrhythmia alarm onset at the end of the five minute.

**MIMIC II Arrhythmia dataset.**[5]  The dataset consists of 6 minutes of data sampled at 125Hz, and we use the data from the 10 seconds interval prior to the alarm onset. The MIMIC II dataset includes 8 electrocardiogram (ECG) leads, namely 'I', 'II', 'III', 'V', 'aVF', 'aVL', 'aVR', and 'MCL1', as well as data from three channels: ABP (Arterial Blood Pressure), PAP (Pulmonary Artery Pressure), and CVP (Central Venous Pressure). MIMIC II-VT is a subset of the MIMIC II Arrhythmia dataset, containing only samples related to VT.

**Ventricular Tachycardia annotated alarms from ICUs (VTaC) dataset.** [6] The Ventricular Tachycardia annotated alarms from ICUs (VTaC) dataset contains a total of 6 minutes of data, sampled at 250Hz. We downsampled the original data to 125Hz and similarly used the data from 10 seconds prior to the alarm onset. This dataset consists of data from 2 ECG channels, arterial blood pressure (ABP), and photoplethysmography (PPG).

**PhysioNet Challenge 2015**[7] The PhysioNet Challenge 2015 dataset contains data of varying lengths from 5 minutes to 6 minutes, sampled at 250Hz. We downsampled the original data to 125Hz and similarly used data from the last 10 seconds interval prior to the alarm onset. In addition to electrocardiogram (ECG) data, this dataset includes ABP (Arterial Blood Pressure) and PPG information. The dataset also covers 5 different types of cardiac arrhythmia alarms, namely: Asystole, Extreme Bradycardia, Extreme Tachycardia, Ventricular Tachycardia, and Ventricular Flutter/Fibrillation.

|  | VTaC | MIMIC II-VT | Challenge'15 |
|---|---|---|---|
| N of samples | 5037 | 2802 | 750 |
| % True | 28.60% | 50.71% | 38.93% |
| Arrhythmia Alarm Types | 1 | 1 | 5 |
| Multi-vendor | Y | N | Y |
| ECG (% events) | 100% | 100% | 100% |
| ABP (% events) | 37% | 94% | 54% |
| PLETH (% events) | 94% | 0% | 83% |

Table 14: Overview of three datasets we used in the experiment part.

---

[3]https://github.com/kweimann/ECG-JEPA/tree/main

[4]https://github.com/PKUDigitalHealth/ECGFounder/tree/master

[5]https://archive.physionet.org/mimic2/

[6]https://physionet.org/content/vtac/1.0/

[7]https://physionet.org/content/challenge-2015/1.0.0/

The overview of each dataset is shown in Table 14. In constructing the pre-train dataset, we extracted unlabeled waveform data corresponding to 17,640 VT alarm events from 1,949 patient records without any expert annotations in **VTaC** dataset. Regarding the fine-tuning task, an 8:1:1 split was applied to the VTaC labeled data for training, validation, and testing respectively. For the MIMIC II-VT dataset, an 8:2 split was used for training and testing, and within the training set, an 8:2 ratio was applied to create the training and validation sets. For the PhysioNet Challenge 2015 dataset, we used the same partitioning method as the MIMIC II dataset. The number of different arrhythmia dataset in the PhysioNet Challenge 2015 dataset and MIMIC II Arrhythmia dataset. are shown in Table 15 and Table 16.

| Alarm types | FALSE | TRUE |
|---|:---:|:---:|
| ASY | 100 | 20 |
| EBR | 45 | 45 |
| ETC | 8 | 131 |
| VTA | 253 | 90 |
| VFB | 52 | 6 |
| Total | 458 | 292 |

Table 15: Detailed distribution of various alarms in the Challenge 2015 dataset. ASY, EBR, ETC, VTA and VFB represent Asystole, Extreme Bradycardia, Extreme Tachycardia, Ventricular Tachycardia, and Ventricular Flutter/Fibrillation.

| Alarm Types | True Alarms | False Alarms | Total Number |
|---|:---:|:---:|:---:|
| Tachycardia | 2262 | 611 | 2873 |
| Bradycardia | 602 | 356 | 958 |
| Asystole | 62 | 882 | 944 |
| Ventricular Tachycardia | 1415 | 1353 | 2768 |
| Ventricular Flutter/Fibrillation | 140 | 327 | 467 |
| Total Number | 4481 | 3529 | 8010 |

Table 16: Detailed distribution of various alarms in theMIMIC II Arrhythmia dataset.

Given that each dataset contains different channels, we processed them to have two ECG channels, one ABP (Arterial Blood Pressure) channel, and one PPG (PLETH) channel. For the ECG channels, we processed them according to the priority which is shown in Table 17.

Due to the different value ranges of different channels, for instance, ECG channels often fall between [-2,2], while ABP channels are between [40,180], we need to process the data to enable deep learning models to perform gradient descent effectively. For the three datasets, we utilized min-max normalization to process the data. Specifically, for each record's channel data, we scaled it to the [0,1] range using its maximum and minimum values:

$$t_n = \frac{t_i - \min(t)}{\max(t) - \min(t)} \tag{19}$$

where $t_n$ denotes normalized channel values, $t_i$ denotes the each values in the channel. $\max(t)$ and $\min(t)$ represent the maximum and minimum values of that channel, respectively.

## F  SLS4Wave in EEG Tasks

Although the SL-S4Wave proposed in this paper is trained on arrhythmia datasets, it can be applied to other long-term medical time series. Similar to ECG, EEG signals in clinical settings have high sampling rates,

| ECG 1 | [ 'II', 'I', 'aVL'] |
|---|---|
| ECG 2 | ['V', 'aVR', 'III', 'aVF', 'MCL', 'V1', 'V2', 'V3', 'V4','V6'] |
| PLETH | ['PLETH'] |
| ABP | ['ABP'] |

Table 17: The channel classifications. The higher the priority, the closer to the front.

multi-channel complexity, and significant environmental noise. However, EEG presents unique challenges, including a greater number of channels (spatial complexity) and the need for more subtle long-term temporal dependencies to distinguish epileptic events from background brain activity. We conducted experiments using SL-S4Wave on three common EEG tasks, including emotion recognition, sleep staging, and mental stress detection, and compared it with common EEG foundation models to demonstrate the effectiveness of our proposed architecture.

### F.1 Task and Dataset

**Pre-Training**. We pretrain SL-S4Wave on the TUH EEG Corpus (Obeid & Picone, 2016). The EEG TUEG dataset usually refers to the Temple University Hospital EEG Corpus (TUEG / TUH EEG Corpus) — a large, publicly available clinical EEG dataset collected from real-world hospital recordings at Temple University Hospital and distributed via the Neural Engineering Data Consortium (NEDC). It is widely used for EEG machine learning research (e.g., seizure detection, abnormal EEG classification, artifact/event detection). In the pretraining process, we use 19-channel EEG. The data are resampled to 200 Hz and divided into 30-second non-overlapping segments.

**SEED-V** (Liu et al., 2021) is an EEG dataset designed for **emotion recognition** [8], covering five emotional categories: *happy*, *sad*, *neutral*, *disgust*, and *fear*. It consists of 62-channel EEG recordings collected at 1000 Hz from 16 subjects, each participating in three sessions. Each session includes 15 trials, which are evenly divided into training, validation, and test sets (5 trials each). The EEG signals are segmented into 1-second windows, yielding a total of 117,744 samples, and resampled to 200 Hz for consistency. The dataset provides rich temporal structure and inter-subject variability, making it a strong benchmark for evaluating generalization in emotion-related EEG modeling.

**ISRUC_S3** (Khalighi et al., 2016) for **sleep stage classification** [9]. The dataset is annotated according to the American Academy of Sleep Medicine (AASM) standard (Berry et al., 2012), with five sleep stages: *Wake*, *NREM1 (N1)*, *NREM (N2)*, *NREM (N3)*, and *REM.* Each EEG segment corresponds to a 30-second epoch. **ISRUC_S3** is a smaller dataset comprising recordings from 10 subjects, also sampled at 200 Hz with six channels, totaling 8,500 labeled segments. We follow an 8:1:1 subject-wise split for training, validation, and testing.

**Mental Arithmetic** dataset (Mumtaz, 2016) supports the task of **mental stress detection** using EEG signals [10]. It contains recordings from 36 subjects under two distinct cognitive conditions: *resting* and *active engagement* in mental arithmetic. EEG data labeled as "no stress" correspond to resting periods prior to the task, while "stress" labels are assigned to recordings during task performance. The signals were acquired using 20 electrodes placed according to the international 10–20 system, with an original sampling rate of 500 Hz. For consistency, the signals are resampled to 200 Hz and band-pass filtered between 0.5–45 Hz to suppress noise. Each recording is segmented into 5-second windows, yielding a total of 1,707 samples.

---

[8] https://bcmi.sjtu.edu.cn/home/seed/seed-v.html

[9] https://sleeptight.isr.uc.pt/

[10] https://figshare.com/articles/dataset/EEG_Data_New/4244171

### F.2 Baselines

**BENDR** (Kostas et al., 2021): We adopted **BENDR (Bert-inspired Neural Data Representations)** as our baseline model, as introduced by Kostas et al. BENDR is a pioneering deep learning architecture for Electroencephalography (EEG) data, leveraging transformers and a contrastive self-supervised learning task. This approach enables the model to learn meaningful representations from vast amounts of unlabeled EEG data.

**BIOT** (Yang et al., 2023): **BIOT (Biosignal Transformer for Cross-data Learning in the Wild)** is a transformer-based architecture designed to handle cross-dataset EEG signal classification under domain shifts. It leverages a domain-invariant attention mechanism and contrastive representation learning to enhance generalization across different recording conditions and subject populations.

**LaBraM**(Jiang et al., 2024): **LaBraM (Large Brain Model)** proposes a scalable transformer-based framework designed to learn generic EEG representations from large-scale brain signal datasets. By pre-training on a diverse corpus of EEG recordings, the model captures rich temporal and spatial features that transfer effectively to various downstream BCI tasks. The architecture incorporates efficient self-attention mechanisms and task-specific adapters to support flexible fine-tuning.

**EEGPT** (Wang et al., 2024): **EEGPT** employs a dual self-supervised learning strategy that combines masked autoencoding with spatial-temporal representation alignment, enhancing feature quality by focusing on high signal-to-noise ratio (SNR) representations rather than raw signals. The model's hierarchical architecture decouples spatial and temporal processing, improving computational efficiency and adaptability to various brain-computer interface (BCI) applications.

### F.3 Metrics

In EEG tasks, we use the following metrics to evaluate performance of model results:

- **Balanced Accuracy**, is a metric used to evaluate the performance of a classification model, specifically designed to handle imbalanced datasets where one class significantly outnumbers the others.

$$Balanced\ Accuracy = \frac{Recall + Specificity}{2} \tag{20}$$

  where

$$Recall = \frac{TP}{TP + FN}, Specificity = \frac{TN}{TN + FP} \tag{21}$$

- **Cohen's Kappa**, which quantifies inter-class agreement beyond chance and is employed as the primary metric for multi-class classification. The Kappa can be calculated by:

$$\kappa = \frac{p_o - p_e}{1 - p_o}, \tag{22}$$

  where $p_o$ denotes the observed agreement or accuracy, and $p_e$ denotes the expected agreement.

- **Weighted F1 Score**, which combines precision and recall while adjusting for class support, ensuring fair performance measurement across imbalanced datasets.

$$weight\ F1 = \sum_{i=1}^{C} w_i \frac{2 * Precision * Recall}{Precision + Recall}, \tag{23}$$

  where $w_i$ is the weight of class $i$.

### F.4 Results and Analysis

To further evaluate the generalization capabilities of SL-S4Wave beyond cardiac signals, we conducted experiments on three diverse EEG benchmarks covering emotion recognition (SEED-V), cognitive load assessment

(Mental Arithmetic), and sleep staging (ISRUC_S3). We compared our framework against EEG foundation models, including BENDR, BIOT, LaBraM, and EEGPT. As shown in Table 18, SL-S4Wave consistently outperforms these Transformer-based baselines across all metrics. The table shows the average results and standard deviation of 5 experiments.

Notably, on the Mental Arithmetic dataset, SL-S4Wave achieves a substantial improvement, surpassing the strongest baseline (LaBraM) by 11.31% in Balanced Accuracy (88.52% vs. 77.21%) and 13.85% in Cohen's Kappa. On the ISRUC_S3 sleep staging task, our method also demonstrates superior performance with a Weighted F1 score of 82.02%, outperforming the runner-up by 3.59%. Even on the challenging SEED-V dataset, SL-S4Wave maintains the leading position. These results indicate that the structured state-space modeling in S4Wave is uniquely advantageous for capturing the complex, non-stationary, and long-range temporal dynamics inherent in multi-channel EEG signals, proving its efficacy as a general-purpose physiological encoder.

Table 18: Performance comparison across EEG datasets.

| Dataset | Model | Balanced Acc | Cohen's Kappa | Weighted F1 |
|---|---|---|---|---|
| SEED-V | BENDR | $22.31 \pm 0.59$ | $3.35 \pm 0.62$ | $20.26 \pm 3.30$ |
| | BIOT | $38.37 \pm 1.87$ | $22.61 \pm 2.62$ | $38.56 \pm 2.03$ |
| | LaBraM | $39.76 \pm 1.38$ | $23.86 \pm 2.09$ | $39.74 \pm 1.11$ |
| | EEGPT | $30.61 \pm 0.62$ | $13.23 \pm 0.52$ | $30.90 \pm 0.44$ |
| | **SL-S4Wave (Ours)** | $\mathbf{40.33 \pm 0.09}$ | $\mathbf{25.79 \pm 0.10}$ | $\mathbf{41.20 \pm 0.08}$ |
| Mental Arithmetic | BENDR | $62.48 \pm 7.65$ | $36.61 \pm 6.72$ | $56.81 \pm 4.48$ |
| | BIOT | $75.36 \pm 1.44$ | $60.04 \pm 1.95$ | $68.75 \pm 1.86$ |
| | LaBraM | $77.21 \pm 0.93$ | $59.99 \pm 1.55$ | $69.09 \pm 1.25$ |
| | EEGPT | $71.62 \pm 1.71$ | $50.81 \pm 2.75$ | $55.97 \pm 1.71$ |
| | **SL-S4Wave (Ours)** | $\mathbf{88.52 \pm 1.22}$ | $\mathbf{73.84 \pm 2.86}$ | $\mathbf{73.07 \pm 0.73}$ |
| ISRUC_S3 | BENDR | $63.52 \pm 0.95$ | $59.95 \pm 1.51$ | $67.89 \pm 1.42$ |
| | BIOT | $75.98 \pm 1.09$ | $71.68 \pm 1.19$ | $78.34 \pm 0.96$ |
| | LaBraM | $76.17 \pm 1.22$ | $71.94 \pm 1.62$ | $78.43 \pm 1.89$ |
| | EEGPT | $66.50 \pm 3.11$ | $61.60 \pm 8.56$ | $63.75 \pm 6.32$ |
| | **SL-S4Wave (Ours)** | $\mathbf{78.56 \pm 0.31}$ | $\mathbf{76.71 \pm 0.91}$ | $\mathbf{82.02 \pm 0.71}$ |

# G  Data Augmentation

For the unlabeled sub-dataset of VTaC, we used the same noise reduction method as described in the VTaC paper Lehman et al. (2024). For ECG, we perform the following filtering: 1) a high-pass filter with 1-Hz cutoff frequency to suppress residual baseline wander; 2) a second-order 30 Hz Butterworth low-pass filter to reduce high frequency noise; and 3) a notch filter to eliminate power line interference. For PPG signal, we utilize a high-pass filter with a stopband frequency of 0.3 Hz and a passband frequency of 0.5 Hz, along with a low-pass filter with a passband frequency of 5 Hz and a stopband frequency of 8 Hz. Figure 7 shows an example of using a filter to eliminate high-frequency noise.

# H  Example ECG Waveforms Prior to Arrhythmia Alarm Onsets

## H.1  True and False Alarm Examples

False alarms in the ICU are commonly triggered by factors like noise, patient movement, lead dislodgement, or incorrect ECG feature recognition by monitoring devices. Bedside monitors may also mistakenly identify different arrhythmia alarms as a specific type of arrhythmia syndrome. Typically, a bedside monitor will sound an alarm within 10 seconds of a severe arrhythmia event. For example, Figure 8 illustrates both a true

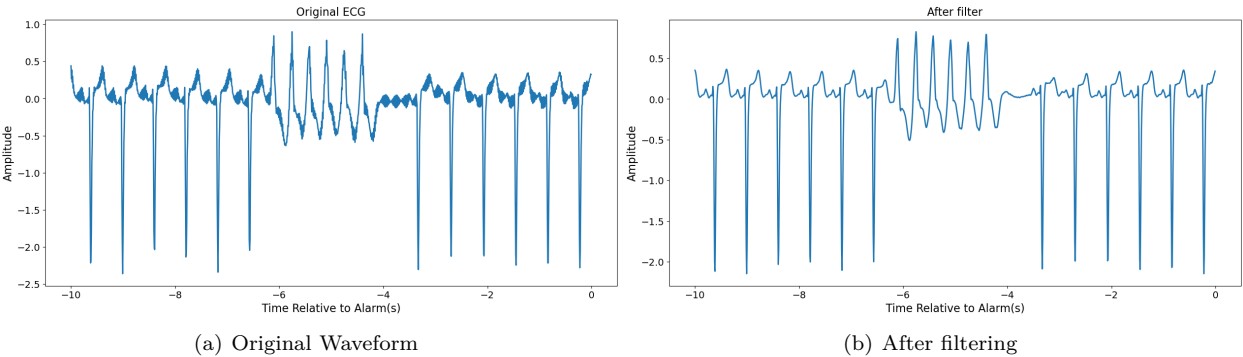

(a) Original Waveform

(b) After filtering

Figure 7: An example of using the filter, the filter eliminates high-frequency noise present in the data.

and false alarm scenario for Ventricular Tachycardia. In this case, the false alarm occurs when the bedside monitor confuses atrial fibrillation with ventricular tachycardia.

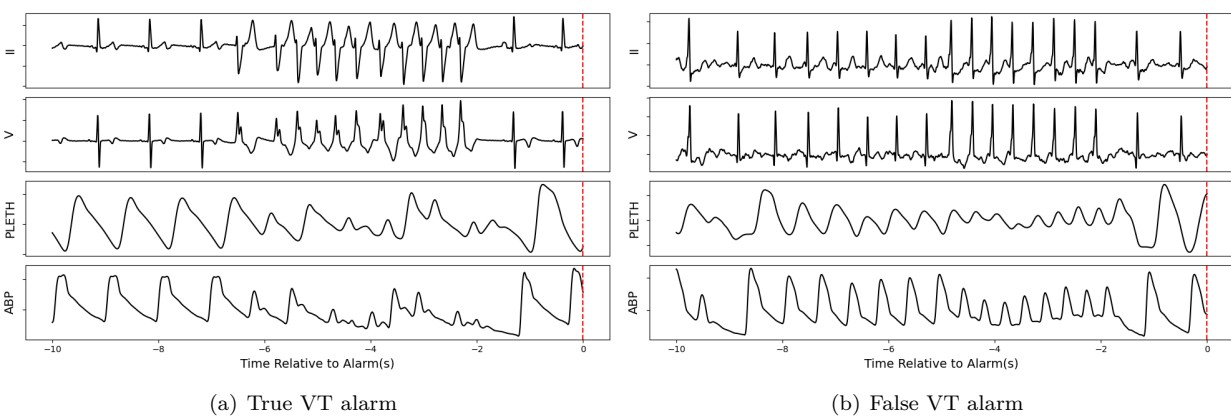

(a) True VT alarm

(b) False VT alarm

Figure 8: Example true vs. false VT alarms. Each plot shows data in the 10-second interval immediately prior to the VT alarm onset. The alarm onset is marked with a vertical red-line at time 0. Figure (b) shows an example false VT alarm – the event corresponds to an episode of atrial fibrillation with rate-related aberration instead of a ventricular tachycardia.

## H.2 Premonition Example: Abnormal Precursor Prior to Alarm Onset

We illustrate in Figure 9 an example where an abnormal waveform pattern appears before the 10-second window immediately preceding the alarm onset. This observation motivates the use of longer temporal segments to improve false alarm classification performance.

The figure shows a real ventricular tachycardia (VT) event. The red dashed line at time $t = 0$ marks the alarm trigger, which is configured to signal a VT event within the preceding 10 seconds (delimited by the black dashed line). The waveform corresponding to the VT episode is highlighted in a red box. The VT begins approximately at $t = -4$ s, lasts for about 3 s, and then returns to a normal rhythm. Notably, around $t = -15$ s, a short arrhythmic episode can be observed (indicated by the black box). Although this segment does not satisfy the clinical definition of VT—typically requiring at least five consecutive ventricular beats with a heart rate exceeding 100 bpm—it exhibits an early ventricular ectopic activity, suggesting a potential precursor to the subsequent VT event.

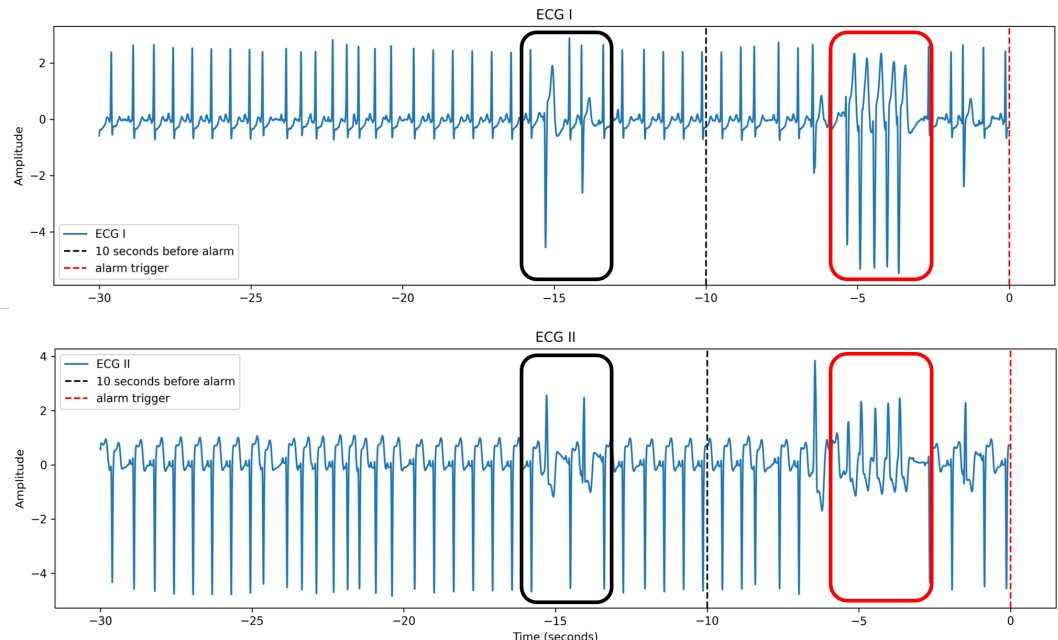

Figure 9: An example of abnormal precursor prior to VT alarm onset. Abnormal waveform pattern appears before the 10-second window immediately preceding the alarm onset.

