# OpenReview forum: "SL-S4Wave: Self-Supervised Learning of Physiological Waveforms with Structured State Space Models"
_TMLR — Accepted by TMLR_

### Review · Reviewer_cMJi · 2026-01-04

**Summary Of Contributions:**

This paper addresses the challenge of modeling long-sequence medical time-series data, where annotated data is costly and scarce, and current methods fail to extract long-term dependencies. To solve these issues, the paper explores a two-stage learning framework based on a contrastive learning loss function and structured state-space models. In the pre-training stage, the authors use a contrastive loss (which contains noise-resilient loss and context consistency loss terms) to conduct self-supervised learning on a specialized state-space model called SL-S4Wave. The purpose of the pre-training stage is to learn a strong feature encoder. In the fine-tuning stage, high-quality annotated data are used to train the downstream task classifier.

Major Strengths:

1. The proposed method dramatically outperforms existing methods, achieving a 49% higher Challenge Score and a 21% higher AUC on the small Challenge 2015-VT dataset.

2. The framework is scalable with respect to sequence length. In contrast to other methods, whose performance generally decreases as sequence length increases, SL-S4Wave can achieve even higher scores with longer sequence windows.

Major Weaknesses:

1. The paper proposes a general framework but only tests it on a single medical task: binary classification for arrhythmia detection. It is difficult to assess whether the proposed framework can be extended to other tasks and achieve similar results. From my perspective, the techniques used—for example, raw signal augmentation, contrastive loss, SGConv, and GELU—are not specifically tailored for arrhythmia detection. If the focus of this paper is to introduce a new method for long-term Physiological Waveforms data, more general experiments are needed. If the experiments mainly focus on this medical task, I would suggest submitting the paper to a more medically oriented journal.

2. The comparison methods are relatively outdated. The most recent baseline was proposed in 2022. Although SL-S4Wave achieves significant improvements, it is unclear how it compares with more recent work. Now it’s hard to evaluate the performance.

3. It would be beneficial to provide more analysis of the S4Wave block to demonstrate that it is not merely a trivial combination of existing neural network sub-blocks. Another option is provide additional ablation studies to show the impact of each component within S4Wave.

**Audience:**

No

**Audience Explanation:**

From current results showed in the experiments, I think the researchers who are working on arrhythmia would be more interested in this work. The TMLR's audience might like the methods or frameworks that seem work for broader type of tasks. For this perspective, at least I am not convinced. Medical-related journals would be better fit for current version of paper.

**Broader Impact Concerns:**

No need

**Claims And Evidence:**

No

**Claims Explanation:**

Since the framework propose in this paper is more like a variant from previous methods and the analysis of the use of each component is less discussed, the strength of this paper highly relies on the experiments. However, as mentioned in previous Summary Of Contributions section, the experiments are limited to one task arrhythmia detection and the baselines are kind of old (the newest among the baselines is proposed in 2022). Then it's hard to evaluate the generality of the method.

**Requested Changes:**

1. more detailed analysis or discussion for the choice of each component inside S4Wave blocks and the contrastive loss terms.
2. test on more tasks. They could be other similar medical sequence task, but just not only focus on arrhythmia detection.
3. compare with more recent baselines.

minor change:
1. add description and structure of Transformer that you used in experiments.

---

> ### Author Response · Authors · 2026-02-26
>
> Thank you very much for your detailed and constructive feedback. Your comments have been very helpful in identifying points that need clarification and have prompted us to make several improvements in the paper, especially in the experimental section. All our changes have been marked in blue in the manuscript. Below are some responses to your valuable suggestions:
>
> >**Need more general task evaluation.**
>
> Thank you for your suggestions and the opportunity to clarify the general capabilities of the proposed model. The contrastive learning framework and model architecture we proposed can be applied to other long-term medical time series. Based on your suggestions, we have added experiments in the **Appendix E** using EEG (electroencephalogram), measuring electrical activity of the brain. We conducted experiments on three sub-tasks: sleep stage detection, emotion recognition, and mental stress detection, comparing performance from using our approach with newer EEG foundation models.
>
> Regarding the concern that the machine learning community may have limited interest in cardiac signals, we would like to clarify that research on ECG and other cardiac waveforms is regularly published at leading machine learning venues, including conferences such as AAAI and ICML [1-3].  In recent years, there has been sustained and growing interest in ECG-based modeling, arrhythmia detection, and physiological time-series learning more broadly. Related work has frequently appeared not only at major ML conferences but also in TMLR and similar venues [4-7]. This continued presence in top-tier venues demonstrates that the machine learning community recognizes both the methodological challenges and the broader scientific importance of this data modality and its associated classification tasks.
>
> Refence:
>
> [1].Kiyasseh, Dani, Tingting Zhu, and David A. Clifton. "CLOCS: Contrastive learning of cardiac signals across space, time, and patients." ICML, 2021.
>
> [2].Lan, Xiang, et al. "Intra-inter subject self-supervised learning for multivariate cardiac signals." AAAI. 2022.
>
> [3].Raghu, Aniruddh, et al. "Sequential multi-dimensional self-supervised learning for clinical time series." International Conference on Machine Learning. PMLR, 2023.
>
> [4].Lalam, Sravan Kumar, et al. "ECG representation learning with multi-modal EHR data." TMLR, (2023).
>
> [5]. Yu, Han, Peikun Guo, and Akane Sano. "ECG semantic integrator (esi): A foundation ecg model pretrained with llm-enhanced cardiological text." TMLR, (2024).
>
> [6]. Upadhyay, Uddeshya, et al. "Hypuc: Hyperfine uncertainty calibration with gradient-boosted corrections for reliable regression on imbalanced electrocardiograms." TMLR, (2023).
>
> [7].Wu, Chenrui, et al. "Efficient Personalized Adaptation for Physiological Signal Foundation Model." ICML. (2025).
>
>
> >**Adding updated baseline**
>
> Thank you for your review feedback. We have added the latest ECG foundation model as a baseline model, including ECG-JEPA [1] and ECGFounder [2]. It can be observed that our model performs better on noisy and small datasets even when compared to the recent ECG foundation models.
>
> [1] Weimann, Kuba, and Tim OF Conrad. "Self-supervised pre-training with joint-embedding predictive architecture boosts ECG classification performance." Computers in Biology and Medicine 196 (2025): 110809.
>
> [2] Li, Jun, et al. "An electrocardiogram foundation model built on over 10 million recordings." Nejm ai 2.7 (2025): AIoa2401033.
>
> >**Adding detailed ablation study**
>
> Thanks for the suggestions! We also recognize our insufficient ablation analysis of the model components. In our previous ablation study, we have already compared the impact of different contrastive learning losses on model performance. We added ablation experiments for model components in the revised **Table 2**. These ablation experiments demonstrated the effectiveness of S4wave.
>
> >**Adding the description and structure of Transformer**
>
> Thank you for catching this, we have included the structure information of the Transformer we used in **Appendix B.3**.

---

> > ### Comment · Reviewer_cMJi · 2026-02-26
> > **Response by Reviewer**
> >
> > Thank you for the detailed response and revisions! All my concerns have been addressed.

---

### Review · Reviewer_BxXP · 2026-01-05

**Summary Of Contributions:**

To overcome the limitation of existing S4-based methods for capturing the information of multichannel physiological
waveforms, the authors develop a new method called SL-S4Wave which introduces the contrastive learning techniques into a tailored S4-based encoder for modeling long-sequence time series data.

**Audience:**

Yes

**Audience Explanation:**

Leveraging constrastive learning in S4-based method to capture various waveform information could be helpful for understanding time series data modeling.

**Claims And Evidence:**

Yes

**Claims Explanation:**

Time series data mining is a hot topic in the field of data mining as well as machine learning.

**Requested Changes:**

- The motivation of introducing the contrastive learning is not very clear. Please highlight the motivation of this module, which is the core part of the proposed method.
- The contributions should be reorganized  in a concise form.
- There are several formatting issues, please carefully proofread the manuscript.
- Please conduct experiments to validate the influence of $\lambda$ on model performance.
- The discussions of the experimental results in Table 1 are very shallow. Given the significant performance gains, please provide deep insights for these results.

---

> ### Author Response · Authors · 2026-02-26
>
> Thank you very much for your detailed and constructive feedback. Your comments are very helpful in improving the clarity and scope of the paper. We have uploaded our revised manuscript, and all the revisions have been marked in blue. Our response to the requested changes you proposed as follows:
>
> >**The motivation of contrastive learning**
>
> Thank you for highlighting this important point. We agree that the motivation for contrastive learning should be made clearer. Contrastive learning is well suited to our setting because it can learn discriminative representations from large-scale unlabeled ECG data, reducing reliance on costly annotations. By constructing positive and negative pairs through augmentations, it provides an effective self-supervised signal.
> Compared with alternatives such as autoencoders or clustering-based methods, contrastive learning more directly encourages invariant and separable representations, which are often more robust and transferable for downstream classification. Prior work on ECG also suggests that contrastive pretraining is especially beneficial when labeled data are limited, although constructing physiologically meaningful pairs remains challenging.
>
> We have revised the Introduction to clarify this motivation and better explain the pair-construction challenge addressed by our method.
>
> >**The contributions should be reorganized in a concise form**
>
> We agree and appreciate this suggestion. We have revised the contribution section to a shorter version.
>
> >**There are several formatting issues in the paper**
>
> Thank you for pointing this out. We will carefully proofread the entire manuscript and correct formatting inconsistencies.
>
> >**The $\lambda$ experiments**
>
> We thank the reviewer for this suggestion. We would like to clarify that an ablation study analyzing the influence of the hyperparameter $\lambda$ has already been conducted and is reported in Appendix B.5. Specifically, Appendix B.5 presents experiments under different values of $\lambda$ and demonstrates that the proposed method is stable across a wide range of $\lambda$, while achieving optimal performance within a reasonable interval. Due to space limitations in the main paper, we placed these detailed ablation results in the appendix. Following the reviewer’s suggestion, we will further highlight this experiment more explicitly in the main text and add a clear pointer to Appendix B.5 in the revised version.
>
> >**Further discussion about the main table.**
>
> Thanks for your great suggestion! The significant performance improvement observed in Table 1 can be attributed to two primary factors.
>
> (1) Effective representation learning from unlabeled data via self-supervised learning. Self-supervised pretraining enables the model to learn informative and transferable representations from large amounts of unlabeled data, resulting in substantially better parameter initialization compared to purely supervised training. This advantage is particularly pronounced in small labeled datasets, where supervised models are prone to overfitting or even optimization instability due to limited training samples. By contrast, self-supervised pretraining provides a strong inductive bias that improves generalization and downstream performance.
>
> (2) The superior capacity of S4Wave to model long time series.
>  While self-supervised learning provides useful prior structure, the backbone architecture ultimately determines how effectively temporal dependencies are captured. Methods such as SimCLR and CMSC rely on CNN-based backbones, which can struggle to efficiently model very long-range dependencies, particularly in high–sampling rate signals. In contrast, architectures with greater expressive capacity, such as SL-S4Wave, TS-TCC, and especially our proposed S4Wave, are better suited for modeling long time horizons.
>
> S4Wave is specifically designed to handle long, high-resolution time series, enabling more effective feature extraction across extended temporal contexts. This architectural advantage has a crucial role in achieving the observed performance gains.

---

> > ### Comment · Reviewer_BxXP · 2026-04-01
> >
> > Thanks for your reply which has addressed my concerns. I have no further questions.

---

### Review · Reviewer_27vA · 2026-03-11

**Summary Of Contributions:**

The paper proposes SL-S4Wave, a combination of a contrastive learning framework for physiological signals and state-space model based architecture. The encoder builds on SGConv with spatial normalization, gating, and residual connections, while the SSL framework uses noise as augmentation and temporal contrastive losses. The evaluation demonstrates strong performance on arrhythmia detection across multiple ECG datasets and additional EEG benchmarks.

**Strength:**
* The evaluation covers extensive baselines.
* Methodology is clear to follow.
* The evaluation is extensive and shows consistent improvements on ablation, cross-domain transferability, and label efficiency.

**Weakness:**
* It is unclear on how the encoder architecture, the SSL framework, and the augmentation strategy could really benefit training. The main results (Table 1) compare SL-S4Wave against baselines that use different encoders, so it is unclear how much improvement comes from the S4Wave architecture versus the proposed contrastive objectives. A modular combination across multiple encoders (CNN, Transformer, proposed) with multiple SSL frameworks (SimCLR, CMSC, proposed) would be needed to properly disentangle these contributions (e.g., adding proposed contrastive learning to existing architectures, and adding S4Wave model to existing contrastive learning methods).
* The augmentation for baselines is unclear and potentially unfair. SL-S4Wave uses domain-specific signal-processing filters (high-pass, low-pass, Butterworth, notch) as augmentations, which is a strong inductive bias for physiological signals. However, the paper does not specify what augmentations were used for SimCLR, CMSC, and other baselines, only referencing default implementations. Contrastive learning is highly sensitive to augmentation quality, and giving baselines the same domain-informed augmentations could close the performance gap substantially.
* The long-sequence advantage appears to come from the state space model architecture, not the contrastive learning framework. Any SSL method paired with S4Wave should similarly benefit from longer inputs. Claiming this as a framework-level contribution is misleading.
* There is no efficiency evaluation despite efficiency being a core motivation. The paper argues that Transformers have quadratic complexity and S4/SGConv is more efficient, but provides no wall-clock training time, inference latency, or memory scaling comparisons.
* Supervised S4Wave already achieves strong performance, which surpasses most SSL baselines in Table 1. This raises questions about the proposed SSL framework, whether a different SSL framework could also achieve a good performance.
* Although the empirical results is impressive, novelty might be weak since CL is simply temporal contrastive with augmentation contrastive learning, and architecture heavily relies on existing works.

**Additional Comments:**

N/A

**Audience:**

Yes

**Audience Explanation:**

The paper studies new model architecture and learning framework for health, which is an important application interested by the TMLR's audience.

**Claims And Evidence:**

No

**Claims Explanation:**

I really enjoyed the paper, and the evaluation is comprehensive. However, it is quiet ambiguous on which component that truly improves the performance. Evaluation suggested in the Summary section could help clarify the work.

**Requested Changes:**

Most of the requested changes match the weakness above. Below are some recommended changes suggested from most critical to least critical:

1. Could the authors provide a modular comparison by combining the proposed S4Wave encoder with existing SSL methods (e.g., SimCLR, CMSC) and conversely applying the proposed contrastive losses to existing encoders (e.g., CNN, Transformer), to disentangle the contributions of the architecture versus the learning framework?
2. What specific augmentations were used for each baseline method, and how would baseline performance change if they were given the same domain-specific signal-processing filters used by SL-S4Wave?
3. Since supervised S4Wave already outperforms most SSL baselines in Table 1, could the authors clarify how much of the overall improvement is attributable to the SSL framework itself versus simply having a stronger encoder architecture?
4. Given that efficiency is a core motivation for adopting state space models over Transformers, could the authors report wall-clock training time, inference latency, and memory usage across methods, especially as sequence length increases from 10s to 30s?

---

> ### Author Response · Authors · 2026-03-21
>
> We sincerely thank the reviewers for their thorough and insightful comments. We have carefully addressed each point below.
>
> > **Question 1:  Disentangle the contributions from the proposed SSL methods vs. the S4Wave encoding architecture.**
>
> Thanks for your suggestion! Following your suggestion, we conducted experiments to disentangle the contributions of the architecture and the learning framework.
>
> Apply the proposed contrastive losses to existing encoders:  In **Appendix B3, Table 6**, we evaluated different model backbones (ResNet, FCN, and Transformer) under our proposed SSL strategy. The results show that most backbones benefit from our SSL framework, except FCN. We hypothesize that this is due to FCN's limited parameter capacity, which restricts its generalization ability.
>
> Combining the proposed S4Wave with existing SSL methods:  Additionally, as shown in **Table 7**, we tested S4Wave pre-trained with different SSL frameworks. Our SSL strategy outperformed several baseline SSL methods, and all SSL methods improved over the original S4Wave, confirming that pre-training consistently benefits performance.
>
> > **Question 2:  Baseline model backbone with same signal-processing.**
>
> We agree that this point might be presented ambiguously in the paper. Thanks for bringing this point. We used signal filtering on all baselines, so their pre-training and fine-tuning were conducted on filtered/denoised data. Note that all methods use the filter mentioned in **Section Appendix Data Augmentation**, so they have the same denoising effect on the pre-training dataset.  The only exception is the rule-based approach which came with its own signal filtering/preprocessing pipeline.
>
> > **Question 3: Ablation study on the SSL and model backbone.**
>
> Thanks for bringing up this interesting question. **Table 2** shows the results from the ablation study of SL-S4Wave using different contrastive losses and model architecture.   We added additional experimental results in updated our ablation study in **Table 2** based on reviewer comments. The “W/o Pre” variant is S4Wave without pretraining weight. Compared with SLS4wave, the Score metric decreased by 16% and 6.5% on the two datasets, respectively. If the noise resilient loss is retained only during pre-training, the Score metric on both datasets decreases by 6% and 8.5%. If only retained the context loss during pre-training, the performance loss on the two datasets is minimal. We also conduct the ablation experiments about each model module. In addition, **Table 6** demonstrates the results about using our SSL framework but replaced with different model backbones.  See also our response to Question 1 above for additional details.
>
> > **Calculation resources.**
>
> We appreciate the reviewers' concern regarding computational resources. We have included a detailed efficiency analysis comparing S4Wave and Transformer in terms of FLOPs, GPU memory, and training speed across 10s, 20s, and 30s sequence lengths in **Appendix B7**, which demonstrates the scalability advantages of S4Wave on longer sequences.

---

### Decision · Action_Editor_GTH7 · 2026-05-01

**Recommendation:** Accept as is

**Audience:**

Yes

**Audience Explanation:**

People working on long-sequence time series, such as ECG and EEG, would be interested in the new model architecture and learning framework proposed in this paper.

**Claims And Evidence:**

Yes

**Claims Explanation:**

The claims are supported by clear and convincing evidence. The paper presents extensive experimental results across multiple ECG and EEG datasets, including strong improvements in key metrics, along with thorough ablation studies and cross-domain evaluations that validate the effectiveness of the proposed framework.